# Sex differences in the association between visceral adiposity index and biological aging: A cross-sectional analysis of NHANES 1999–2018 with mediation by insulin resistance

Jia Yang[1◦], Haifeng Liu[1*◦], Xupeng Huang[1], Zimin Fu[1], Jie Zhou[1], Tiejun Liu[2], Weimin Zhao[3]

**1** College of Chinese Medicine, Changchun University of Chinese Medicine, Changchun City, Jilin Province, China, **2** Department of Gastroenterology, First Affiliated Hospital to Changchun University of Chinese Medicine, Changchun City, Jilin Province, China, **3** Department of Neurology, First Affiliated Hospital to Changchun University of Chinese Medicine, Changchun City, Jilin Province, China

◉ These authors contributed equally to this work.
* 18447059019@163.com

## Abstract

### Background

Aging poses challenges global health and social care systems, with obesity potentially being associated with this process. However, sex differences and the mediating mechanisms underlying this association remain poorly understood. This study investigated sex differences in the associations of visceral adiposity index (VAI) with biological aging (BA) assessed via the Klemera-Doubal method age (KDMAge) and the risk of KDMAge acceleration (KDMAgeAccel), and explored the mediating role of the homeostasis model assessment of insulin resistance (HOMA-IR).

### Methods

According to the National Health and Nutrition Examination Survey (1999–2018), weighted data from 19486 adults were analyzed cross-sectionally. VAI–BA associations were analyzed in the overall population and sex-specific subgroups using multivariable linear and logistic regressions, with nonlinear patterns explored via restricted cubic splines (RCSs) and threshold analyses. Mediation analyses quantified the mediating effects of HOMA-IR.

### Results

VAI correlated positively with BA overall. However, females showed stronger associations than males for each 1-unit VAI increase with KDMAge elevation ($\beta_{female}$ = 1.02, 95% CI: 0.81–1.23; $\beta_{male}$ = 0.59, 95% CI: 0.42–0.75) and KDMAgeAccel risk ($OR_{female}$ = 1.22, 95% CI: 1.17–1.28; $OR_{male}$ = 1.10, 95% CI: 1.07–1.13), with all

**Data availability statement:** The study dataset "DATA.xlsx (3.16 MB)" has been uploaded and is hosted on Figshare with the DOI: https://doi.org/10.6084/m9.figshare.30153427.

**Funding:** The author(s) received no specific funding for this work.

**Competing interests:** The authors have declared that no competing interests exist.

associations remaining significant ($P < 0.001$). RCSs demonstrated nonlinear positive associations in all cohorts ($P_{nonlinear} < 0.001$). Threshold analyses identified that females exhibited a higher VAI threshold for the KDMAge association (3.52 vs. 2.60) despite comparable effect sizes, whereas similar VAI thresholds across sexes were associated with a greater KDMAgeAccel risk in females than in males (75% vs. 36%). HOMA-IR partially mediated VAI–BA associations, with a greater percentage in males than in females in both VAI–KDMAge (21.67% vs. 12.71%) and VAI–KDMAgeAccel risk (27.09% vs. 10.39%) relationships.

## Conclusion

VAI showed a significant positive association with BA, with females demonstrating a stronger association strength and males showing a greater proportion of the association mediated through HOMA-IR.

---

## Introduction

The global population is rapidly aging [1]. Epidemiological projections indicate that individuals aged ≥ 60 years will constitute 16% of the global population by 2030, rising to 22% (2.1 billion) by mid-century [2]. Aging is robustly linked to increased risks of multiple chronic conditions, including diabetes mellitus (DM), cardiovascular disease (CVD), hypertension (HTN), chronic kidney disease (CKD), and cancer [3–5]. Given the profound impact of these conditions on healthcare burdens and age-related mortality, accurately assessing biological aging (BA) is crucial for developing interventions to decelerate it. Biological age, a core indicator of physiological decline, provides superior predictive value over chronological age [6]. The Klemera-Doubal method (KDMAge) quantifies BA by integrating multisystem biomarkers (e.g., metabolic, inflammatory, cardiovascular) [7]. Unlike single biomarkers, KDMAge captures individual heterogeneity in aging trajectories through multi-system interactions, serving as an effective predictor of disease risk, functional decline, and mortality [8].

The global overweight/obesity epidemic poses a major public health threat, with prevalence rising exponentially [9]. Obesity independently predicts CVD, HTN, DM, and cancer, ranking as the second leading modifiable mortality risk factor after tobacco in Western populations [10]. In the United States, 70% of adults and 19% of adolescents are affected [11]. Significant sex-based differences exist: females show higher obesity prevalence (18% vs. 14% in males in 2020) with projections indicating persistent disparities (27% vs. 23% by 2035). Adipose distribution also differs markedly, with gluteofemoral predominance in females versus abdominal in males, influenced by biological (e.g., sex hormones) and sociocultural factors [12]. Given these health implications, effective obesity assessment is critical. While body mass index (BMI) and waist circumference (WC) provide unidimensional measures, the visceral adiposity index (VAI) offers a sex-specific metric integrating morphological (BMI, WC) and biochemical markers (triglycerides, HDL). This enables simultaneous

evaluation of visceral fat accumulation and metabolic dysfunction, establishing VAI as a validated clinical marker for visceral obesity [13].

Obesity accelerates aging through pathways including vascular dysfunction, insulin resistance (IR), chronic inflammation, and lifestyle factors [14,15]. Of these, IR—clinically measured by homeostatic model assessment (HOMA-IR)—is particularly critical. Obesity impairs insulin signaling via chronic inflammation, lipotoxicity, and metabolic dysregulation [16], while the synergy of obesity and IR promotes age-related comorbidities (e.g., DM, CVD) that exacerbate systemic frailty [17]. However, sex-specific mechanisms mediating the obesity–aging association remain poorly understood, particularly regarding visceral adiposity and IR pathways. Leveraging the National Health and Nutrition Examination Survey (NHANES), the purpose of our investigation was to systematically examine sex-specific associations between VAI and BA while quantifying the mediating role of IR, thereby offering novel insights to inform public health strategies aimed at mitigating obesity-related aging risks.

## Materials and methods

### Data source and study population

We analyzed deidentified NHANES data (1999–2018) through retrospective cross-sectional methods following program access protocols (https://wwwn.cdc.gov/nchs/nhanes/). Publicly available NHANES data were analyzed, collected through multi-stage probabilistic sampling combining interviews, physical exams, and mobile unit laboratory assessments. Consequently, these data are nationally representative. The study procedures received approval from the National Center for Health Statistics Research Ethics Review Board, and informed written consent was obtained from participants before data collection began (https://www.cdc.gov/nchs/nhanes/about/erb.html). Given that this secondary analysis exclusively employed aggregated, deidentified data containing no protected health information, supplementary ethical review was not necessary.

In the initial population, exclusions were performed sequentially. First, individuals aged < 20 years were excluded (n = 46235). Second, individuals who were pregnant were excluded (n = 1442). Third, those with incomplete measurements of VAI (n = 31655) and KDMAge (n = 136) were excluded, and fourth, individuals with incomplete dates of HOMA-IR were excluded (n = 223). Finally, covariates without incomplete information were excluded (n = 2139). The analytical cohort included 19486 adults (49.94% female; 50.06% male), representing a nationally representative NHANES sample across two decades. The detailed exclusion criteria and participant flow are illustrated in Fig 1.

### Assessment of VAI and IR

As a sex-specific metric integrating anthropometric and biochemical parameters, VAI was utilized to estimate visceral fat distribution. For female participants, VAI was calculated as follows:

$$VAI_{female} = \frac{WC(cm)}{36.58 + 1.89 \times BMI(kg/m^2)} \times \frac{TG(mmol/L)}{0.81} \times \frac{1.52}{HDL(mmol/L)}$$

While the male-specific formula incorporates adjusted coefficients:

$$VAI_{male} = \frac{WC\ (cm)}{39.68 + 1.88 \times BMI\ (kg/m^2)} \times \frac{TG\ (mmol/L)}{1.03} \times \frac{1.31}{HDL\ (mmol/L)}$$

For this formula, TG and HDL were measured via standardized laboratory protocols (https://wwwn.cdc.gov/nchs/nhanes/search/DataPage.aspx?Component=Laboratory). All variables adhered to the NHANES quality control criteria. IR was assessed by HOMA-IR computed from fasting plasma glucose (FPG) levels as well as fasting serum insulin (FINS) levels

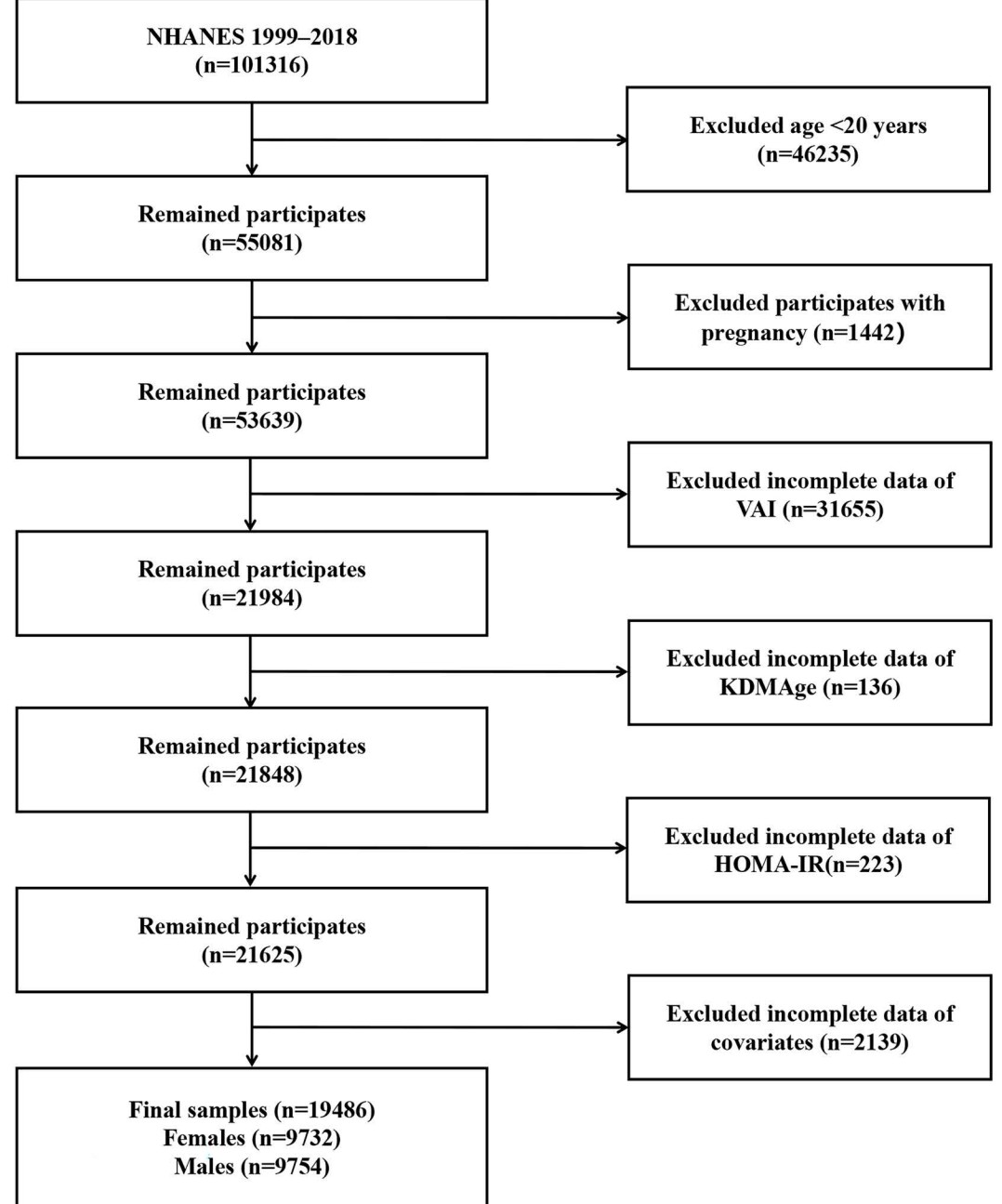

**Fig 1. Flow chart.** VAI, visceral adiposity index; NHANES, National Health and Nutrition Examination Survey; KDMAge, Klemera-Doubal method age; HOMA-IR, homeostasis model assessment of insulin resistance.

[18]. This biochemical index has been widely adopted in clinical settings as the most reliable surrogate marker for quantifying IR, particularly in large-scale epidemiological investigations [19]. HOMA-IR value as per the following formula:

$$HOMA-IR = \frac{FPG\ (mmol/L) \times FINS\ (\mu U/mL)}{22.5}$$

## Assessment of biological aging

BA was assessed via the Klemera-Doubal method age (KDMAge) and KDMAge acceleration (KDMAgeAccel) risk [20]. Sex-stratified parameter optimization was conducted to address sex-specific aging patterns. Ten biomarkers previously validated in aging research were integrated into the KDMAge model: lymphocyte percentage, systolic blood pressure, total cholesterol, serum albumin, blood urea nitrogen, creatinine, mean corpuscular volume, glycated hemoglobin, leukocyte count, and alkaline phosphatase, following established computational frameworks [7]. The computational algorithm is detailed in S1 Table. KDMAgeAccel was derived by subtracting chronological age (CA) from the KDMAge values. Positive residuals (KDMAge > CA) indicate accelerated BA, whereas a nonpositive value suggests no acceleration [21]. The continuous measure was dichotomized to facilitate categorical comparisons of KDMAgeAccel risk in subsequent analyses.

## Assessment of covariates

In this study, potential confounders were systematically adjusted using established demographic, lifestyle, and clinical covariates. Demographics comprised sex (female/male), age groups (20–39/40–59/ ≥ 60 years), race (Mexican American/ non-Hispanic White/non-Hispanic Black/other), education (<high school/high school/ >high school), poverty status defined by the poverty-income ratio (PIR; < 1 indicating below the poverty threshold), and marital status (married or living with partner/divorced or separated or widowed/never married) [22]. Lifestyle assessments included smoking status (never/former/ now), alcohol consumption (yes/no) [23], and physical activity. The Global Physical Activity Questionnaire was employed to quantify physical activity [24]. To minimize longitudinal survey variability, as referenced in previous studies [25], a binary exposure variable was derived: engagement in moderate/vigorous physical activity (M/VPA, ≥ 10-min continuous bouts) versus none, which is consistent with the Global Physical Activity Questionnaire analytic guidelines. Clinical covariates are self-reported or objectively measured comorbidities, including HTN, CVD, DM, CKD, and cancer. Specifically, HTN is defined as systolic blood pressure > 140 mmHg or diastolic blood pressure > 90 mmHg or physician diagnosis; CVD encompasses congestive heart failure, coronary artery disease, myocardial infarction, angina pectoris, or stroke; DM requires glycated hemoglobin ≥ 6.5%, fasting glucose ≥ 126 mg/dL, 2-h oral glucose tolerance test ≥ 200 mg/dL, antidiabetic use, or physician confirmation; CKD is identified by estimated glomerular filtration rate < 60 mL/min/1.73m² (CKD-EPI equation) or urine albumin-creatinine ratio > 30 mg/g; cancer is based on self-reported physician-diagnosed malignancy, see S2 Table for details.

## Statistical analysis

Consistent with NHANES's survey design methodology, analyses accounted for clustering (primary sampling units), stratification variables, and individual sampling weights. Weighted estimates for the national population were derived utilizing the "survey" package in R. The baseline characteristics were stratified by sex. Continuous variables were assessed for normality using Kolmogorov-Smirnov tests; normally distributed variables were analyzed by weighted t-tests and expressed by mean ± standard errors. In contrast, weighted Mann-Whitney tests were utilized for those continuous variables that were determined to be nonnormally distributed, and the variables were reported as the median (1st quartile, 3rd quartile). Categorical variables were compared through weighted chi-square tests presented as unweighted counts with weighted proportions.

Analyses of multivariable linear and logistic regression were performed across three distinct cohorts, namely, the whole population, females, and males, to explore the correlations of VAI with BA (KDMAge and KDMAgeAccel risk). Initially, sex interaction effects were evaluated through stratified analyses and multiplicative interaction terms. VAI was analyzed both continuously and categorically (quartiles) in all models. These analyses utilized three progressively adjusted models. The base model (Model 1, M 1) examined crude associations without covariate adjustment. Model 2 (M 2) was adjusted for age groups, race, educational attainment, status of marriage and poverty, along with a separate adjustment for sex, to

analyze the whole population. The fully adjusted Model 3 (M 3) extended these adjustments by incorporating lifestyle and clinical covariates, including smoking, alcohol intake, M/VPA, HTN, CVD, cancer, and CKD. Odds ratio (OR) as well as regression coefficient (β) and their 95% confidence intervals (CIs) were calculated.

To comprehensively characterize potential nonlinear associations between VAI and BA, we fitted restricted cubic spline (RCS) analyses across three strata: the whole population, the female subgroup, and the male subgroup [26]. This spline-based approach enables flexible visualization of dose-response patterns. The RCS analyses incorporated full covariate adjustments equivalent to Model 3 specifications, with VAI modeled as a continuous exposure. The models of two-piecewise regression were utilized to examine potential threshold effects. We employed likelihood ratio tests to compare piecewise and linear models. This process enabled the determination of the optimal characterization of the dose-response relationships.

Mediation analyses, utilizing a covariate structure like Model 3, were carried out to evaluate HOMA-IR's mediating role in the VAI–BA association. These analyses were conducted across the whole population and sex-stratified subgroups. The effects in mediation analyses were decomposed into direct, indirect, and total effects, with significance determined via nonparametric bootstrap resampling (1000 iterations; random seed = 1234). Effect estimates and 95% CIs were derived from the bootstrap distributions. The proportion of mediated effects was subsequently calculated to evaluate the relative contribution of IR to the observed associations.

Sensitivity analyses were conducted using multiple combinations of covariates to assess the robustness of the observed findings. It is noteworthy that DM was excluded from the primary adjustments to circumvent overadjustment bias, given its potential mediation of the VAI–BA association (S3 Table). Furthermore, HDL is a common component of both VAI and KDMAge metrics, which may lead to potential formulaic confounding. We performed sensitivity analyses further adjusting for DM and serum HDL levels based on Model 3, re-evaluating associations in the whole and sex-stratified populations. In addition, given the strong association between diabetes and both insulin resistance and biological aging, a sensitivity analysis excluding individuals with diagnosed diabetes could help evaluate whether the observed associations are independent of overt diabetic status. These procedures enabled the validation of consistency for the associations in the primary analyses.

To address potential effect modification by demographic factors, we conducted comprehensive subgroup and interaction analyses. Given the racial/ethnic diversity of NHANES participants (Mexican American, Non-Hispanic White, Non-Hispanic Black, Other), we (a) performed race/ethnicity-stratified analyses of VAI–IR, IR–BA and VAI–BA associations; (b) and tested multiplicative interaction terms (VAI × race/ethnicity) in fully adjusted models. Additionally, since age may modify relationships between VAI and BA, we (a) assessed age interaction effects through VAI × age terms; (b) and conducted age-stratified analyses using predefined categories (20–39, 40–59, ≥ 60 years). These analyses were implemented in both the whole cohort and sex-specific subgroups using multivariable-adjusted models. This approach enhances generalizability while identifying population-specific patterns in the visceral adiposity-aging relationship.

Full statistical workflows were performed in R (version 4.3.1). The Zstats 1.0 platform (www.zstats.net) was utilized to increase computational efficiency through automated table generation and graphical outputs [27]. Statistical significance was set at $P < 0.05$ (two-tailed). To ensure analytical accuracy, all the models underwent rigorous verification, including code replication by independent analysts and consistency checks across software environments.

## Results

### Baseline characteristics

The final cohort comprised 19486 participants (9732 females and 9754 males) after rigorous inclusion/exclusion screening. Table 1 summarizes sex-stratified participant characteristics. Pronounced sex disparities were evident in the demographic and lifestyle variables ($P < 0.001$). There were greater proportions of females in the ≥ 60 years age group (25.67% vs. 22.15%), > high school education level (59.89% vs. 56.86%), divorced/separated/widowed status (23.32% vs. 12.27%), poverty rates (14.98% vs. 11.47%), and never-smoking (59.08% vs. 45.40%). Males demonstrated elevated Mexican

**Table 1. Participant baseline characteristics.**

| Variable | Whole population (n = 19486) | Females (n = 9732) | Males (n = 9754) | P-value |
|---|---|---|---|---|
| Age (years) % | | | | <0.001 |
| 20–39 | 6358 (37.08) | 3148 (35.91) | 3210 (38.26) | |
| 40–59 | 6527 (39.00) | 3298 (38.42) | 3229 (39.58) | |
| ≥ 60 | 6601 (23.93) | 3286 (25.67) | 3315 (22.15) | |
| Race % | | | | <0.001 |
| Mexican American | 3357 (7.67) | 1641 (6.93) | 1716 (8.43) | |
| Non-Hispanic White | 8996 (70.57) | 4420 (70.29) | 4576 (70.85) | |
| Non-Hispanic Black | 3803 (10.08) | 1966 (10.96) | 1837 (9.19) | |
| Other | 3330 (11.68) | 1705 (11.82) | 1625 (11.53) | |
| Education % | | | | <0.001 |
| <High school | 5058 (17.18) | 2380 (16.24) | 2678 (18.13) | |
| High school | 4488 (24.43) | 2173 (23.87) | 2315(25.01) | |
| > High school | 9940 (58.39) | 5179 (59.89) | 4761 (56.86) | |
| Marital status % | | | | <0.001 |
| Never married | 3249 (16.56) | 1554 (14.66) | 1695 (18.50) | |
| Divorced/Separated/Widowed | 4243 (17.84) | 2762 (23.32) | 1481 (12.27) | |
| Married/Living with partner | 11994 (65.60) | 5416 (62.03) | 6578 (69.23) | |
| Poverty % | 3821 (13.24) | 2087 (14.98) | 1734 (11.47) | <0.001 |
| Smoking status % | | | | <0.001 |
| Never | 10392 (52.29) | 6110 (59.08) | 4282 (45.40) | |
| Former | 4988 (25.69) | 1912 (21.50) | 3076 (29.96) | |
| Now | 4106 (22.01) | 1710 (19.43) | 2396 (24.65) | |
| Drinking status % | 11828 (65.43) | 4854 (56.95) | 6974 (74.06) | <0.001 |
| M/VPA % | 9391 (55.01) | 4184 (50.08) | 5207 (60.02) | <0.001 |
| HTN % | 7883 (34.74) | 3945 (34.89) | 3938 (34.59) | 0.727 |
| CVD % | 2124 (8.38) | 893 (7.32) | 1231 (9.47) | <0.001 |
| Cancer % | 1789 (8.95) | 950 (10.30) | 839 (7.58) | <0.001 |
| CKD % | 3476 (13.24) | 1820 (14.85) | 1656 (11.59) | <0.001 |
| DM % | 3708 (13.33) | 1746 (12.62) | 1962 (14.05) | 0.008 |
| HDL (mmol/L) | 1.31 (1.09, 1.60) | 1.45 (1.22, 1.73) | 1.19 (1.01, 1.40) | <0.001 |
| VAI | 1.50 (0.91, 2.54) | 1.49 (0.93, 2.50) | 1.51 (0.89, 2.59) | 0.860 |
| HOMA-IR | 2.29 (1.43, 3.92) | 2.15 (1.37, 3.68) | 2.44 (1.50, 4.14) | <0.001 |
| KDMAge (years) | 39.03 (28.26, 52.30) | 40.76 (29.82, 53.82) | 37.42 (26.63, 50.65) | <0.001 |
| KDMAgeAccel (years) | −5.79 (−14.42, 3.34) | −4.97 (−13.03, 3.42) | −6.85 (−16.20, 3.23) | <0.001 |
| KDMAgeAccel risk % | 6391 (33.34) | 3253 (34.46) | 3138 (32.19) | 0.011 |

Nonnormal continuous variables were assessed via weighted Mann-Whitney tests and reported with the median (1st quartile, 3rd quartile). Weighted χ² tests compared categorical variables which were represented as unweighted counts and weighted proportions. M/VPA, moderate or vigorous physical activity; HTN, hypertension; CKD, chronic kidney disease; DM, diabetes mellitus; HDL, high-density lipoprotein; VAI, visceral adiposity index; CVD, cardiovascular disease; HOMA-IR, homeostasis model assessment of insulin resistance; KDMAge, Klemera-Doubal method age; KDMAgeAccel: KDMAge acceleration.

American representation (8.43% vs. 6.93%), alcohol consumption (74.06% vs. 56.95%), and M/VPA (60.02% vs. 50.08%). The incidence of HTN was comparable between sexes (34.89% vs. 34.59%, P = 0.727). However, females had higher rates of cancer (10.30% vs. 7.58%) and CKD (14.85% vs. 11.59%), while males had higher rates of CVD (9.47% vs. 7.32%) and DM (14.05% vs. 12.62%), these differences were all statistically significant (P < 0.01).

No sex-based difference in VAI was detected ($P=0.860$). HOMA – IR was higher in males (2.44 vs. 2.15; $P<0.001$), despite females demonstrating older KDMAge (40.76 vs. 37.42 years; $P<0.001$) and higher KDMAgeAccel risk (34.46% vs. 32.19%; $P=0.011$). As shown in Fig 2, analyses stratified by VAI quartiles (expressed by the median) revealed that both sexes presented progressively elevated CA, KDMAge, KDMAgeAccel and KDMAgeAccel risk ($P<0.001$) across ascending VAI quartiles (Q1–Q4). See S4 Table for details.

## Association between VAI and biological aging

**Multivariable regression analysis.** Multivariable linear and logistic regression analyses (Fig 3A) showed significant positive correlations between the VAI and BA (KDMAge and KDMAgeAccel risk) in the whole population and sex-specific cohorts. In fully adjusted models (M 3) for the whole population, each 1-unit increase in VAI, KDMAge increased by 0.75 years ($\beta=0.75$, 95% CI 0.62–0.88), and KDMAgeAccel risk rose by 14% (OR = 1.14, 95% CI 1.11–1.17). Those in the highest VAI quartile (Q4) exhibited a 6.71-year increase in KDMAge ($\beta=6.71$, 95% CI: 5.90–7.51) and a 2.71-fold greater probability of KDMAgeAccel (OR = 2.71, 95% CI: 2.36–3.10) compared to the participants in the lowest quartile (Q1). All reported associations reached $P<0.001$ significance. These findings confirmed VAI's predictive utility as a continuous and categorical measure.

Sex-stratified analyses (Fig 3B) revealed persistent differential associations between VAI and BA across progressively adjusted models (M 1 to M 3), with statistically significant interaction effects ($P_{interaction}<0.05$). All models showed stronger VAI–BA relationships in females than in males. The absence of interval overlaps between sexes underscores sex-specific disparities. In Fig 3A, females exhibited a greater KDMAge increase per 1-unit increase in VAI ($\beta=1.02$, 95% CI: 0.81–1.23) and a higher risk of KDMAgeAccel (OR = 1.22, 95% CI: 1.17–1.28) compared with males ($\beta=0.59$, 95% CI: 0.42–0.75; OR = 1.10, 95% CI: 1.07–1.13). Quartile-based comparisons further highlighted sex disparities: compared with Q1, Q4 females had a 7.76-year KDMAge increase ($\beta=7.76$, 95% CI: 6.90–8.62) and 3.57-fold accelerated aging odds (OR = 3.57, 95% CI: 2.99–4.25), whereas Q4 males displayed a 6.34-year KDMAge increase ($\beta=6.34$, 95% CI: 5.17–7.52) and 2.39-fold accelerated aging odds (OR = 2.39, 95% CI: 1.97–2.90) compared to Q1. Every one of the correlations mentioned above reached significance at the level of $P<0.001$. All the models (M1 to M3) maintained consistent trends ($P_{trend}<0.001$). The results underscored VAI's different association with BA across sexes.

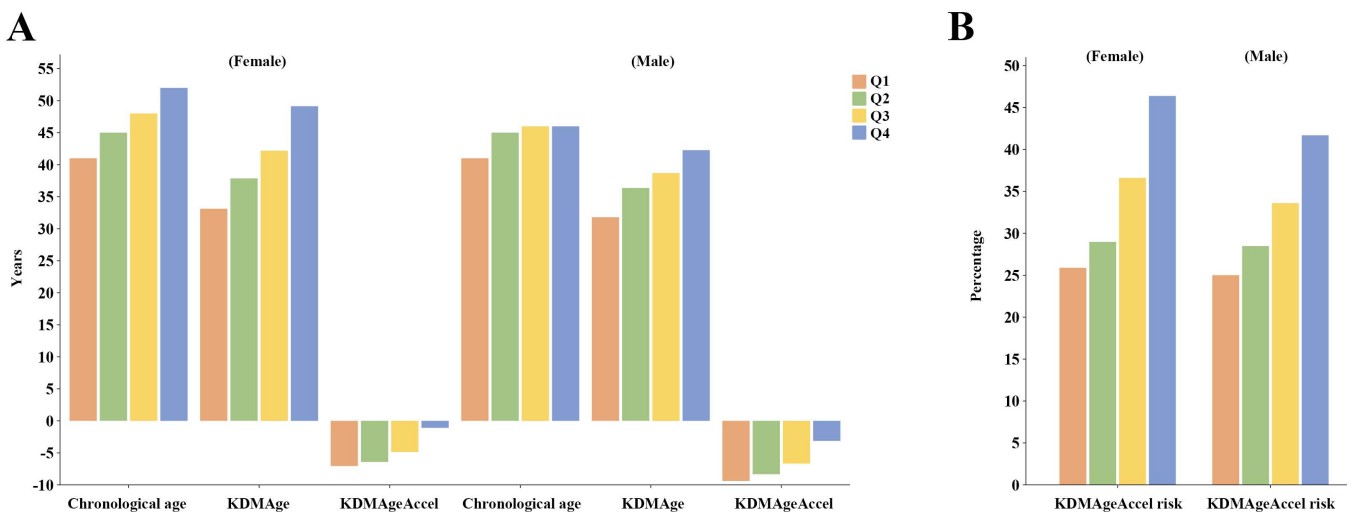

**Fig 2. Sex-specific associations of VAI quartiles with biological aging.** Both sexes exhibited progressively elevated chronological age, KDMAge, KDMAgeAccel (A), and KDMAgeAccel risk (**B**) across ascending VAI quartiles (Q1–Q4). VAI, visceral adiposity index; KDMAge, Klemera-Doubal method age; KDMAgeAccel, KDMAge acceleration.

**A**

| | Variable | KDMAge β (95% CI) | P-value | P-trend | KDMAgeAccel risk OR (95% CI) | P-value | P-trend |
|---|---|---|---|---|---|---|---|
| **Whole** | VAI–M 1 | 1.25 (1.03–1.47) | <0.001 | <0.001 | 1.13 (1.10–1.15) | <0.001 | <0.001 |
| | **Q1** | 0 (Reference) | | | 1 (Reference) | | |
| | Q2 | 3.95 (3.12–4.78) | <0.001 | | 1.16 (1.03–1.31) | 0.013 | |
| | Q3 | 7.69 (6.72–8.66) | <0.001 | | 1.57 (1.37–1.80) | <0.001 | |
| | Q4 | 12.55 (11.57–13.53) | <0.001 | | 2.26 (2.02–2.53) | <0.001 | |
| | VAI–M 2 | 1.00 (0.83–1.16) | <0.001 | <0.001 | 1.17 (1.14–1.20) | <0.001 | <0.001 |
| | **Q1** | 0 (Reference) | | | 1 (Reference) | | |
| | Q2 | 2.07 (1.45–2.69) | <0.001 | | 1.34 (1.17–1.52) | <0.001 | |
| | Q3 | 5.02 (4.26–5.78) | <0.001 | | 2.02 (1.74–2.34) | <0.001 | |
| | Q4 | 8.95 (8.21–9.69) | <0.001 | | 3.24 (2.87–3.67) | <0.001 | |
| | VAI–M 3 | 0.75 (0.62–0.88) | <0.001 | <0.001 | 1.14 (1.11–1.17) | <0.001 | <0.001 |
| | **Q1** | 0 (Reference) | | | 1 (Reference) | | |
| | Q2 | 1.46 (0.83–2.1) | <0.001 | | 1.26 (1.10–1.44) | 0.001 | |
| | Q3 | 3.67 (2.97–4.38) | <0.001 | | 1.80 (1.54–2.09) | <0.001 | |
| | Q4 | 6.71 (5.9–7.51) | <0.001 | | 2.71 (2.36–3.10) | <0.001 | |
| **Female** | VAI–M 1 | 1.74 (1.32–2.16) | <0.001 | <0.001 | 1.17 (1.13–1.22) | <0.001 | <0.001 |
| | **Q1** | 0 (Reference) | | | 1 (Reference) | | |
| | Q2 | 4.1 (2.96–5.24) | <0.001 | | 1.18 (1.01–1.39) | 0.044 | |
| | Q3 | 8.71 (7.45–9.96) | <0.001 | | 1.68 (1.42–1.98) | <0.001 | |
| | Q4 | 14.66 (13.46–15.86) | <0.001 | | 2.49 (2.14–2.89) | <0.001 | |
| | VAI–M 2 | 1.23 (0.94–1.52) | <0.001 | <0.001 | 1.26 (1.21–1.31) | <0.001 | <0.001 |
| | **Q1** | 0 (Reference) | | | 1 (Reference) | | |
| | Q2 | 1.94 (1.18–2.71) | <0.001 | | 1.40 (1.17–1.68) | <0.001 | |
| | Q3 | 5.15 (4.22–6.08) | <0.001 | | 2.35 (1.94–2.85) | <0.001 | |
| | Q4 | 9.55 (8.66–10.44) | <0.001 | | 4.09 (3.41–4.91) | <0.001 | |
| | VAI–M 3 | 1.02 (0.81–1.23) | <0.001 | <0.001 | 1.22 (1.17–1.28) | <0.001 | <0.001 |
| | **Q1** | 0 (Reference) | | | 1 (Reference) | | |
| | Q2 | 1.52 (0.8–2.25) | <0.001 | | 1.34 (1.11–1.61) | 0.002 | |
| | Q3 | 4.2 (3.35–5.05) | <0.001 | | 2.19 (1.80–2.67) | <0.001 | |
| | Q4 | 7.75 (6.86–8.64) | <0.001 | | 3.58 (2.98–4.31) | <0.001 | |
| **Male** | VAI–M 1 | 0.94 (0.72–1.17) | <0.001 | <0.001 | 1.10 (1.06–1.13) | <0.001 | <0.001 |
| | **Q1** | 0 (Reference) | | | 1 (Reference) | | |
| | Q2 | 3.66 (2.51–4.81) | <0.001 | | 1.14 (0.96–1.35) | 0.126 | |
| | Q3 | 6.55 (5.26–7.85) | <0.001 | | 1.46 (1.21–1.77) | <0.001 | |
| | Q4 | 10.52 (9.13–11.9) | <0.001 | | 2.06 (1.74–2.44) | <0.001 | |
| | VAI–M 2 | 0.86 (0.66–1.06) | <0.001 | <0.001 | 1.13 (1.09–1.16) | <0.001 | <0.001 |
| | **Q1** | 0 (Reference) | | | 1 (Reference) | | |
| | Q2 | 2.36 (1.38–3.33) | <0.001 | | 1.31 (1.10–1.56) | 0.003 | |
| | Q3 | 5.09 (3.91–6.27) | <0.001 | | 1.81 (1.47–2.21) | <0.001 | |
| | Q4 | 8.67 (7.56–9.79) | <0.001 | | 2.77 (2.32–3.31) | <0.001 | |
| | VAI–M 3 | 0.59 (0.42–0.75) | <0.001 | <0.001 | 1.10 (1.07–1.13) | <0.001 | <0.001 |
| | **Q1** | 0 (Reference) | | | 1 (Reference) | | |
| | Q2 | 1.64 (0.67–2.62) | 0.001 | | 1.24 (1.03–1.48) | 0.023 | |
| | Q3 | 3.42 (2.3–4.53) | <0.001 | | 1.57 (1.27–1.93) | <0.001 | |
| | Q4 | 6.14 (4.96–7.32) | <0.001 | | 2.31 (1.90–2.80) | <0.001 | |

**B**

| | Variable | KDMAge β (95% CI) | P-value | P-interaction | KDMAgeAccel risk OR (95% CI) | P-value | P-interaction |
|---|---|---|---|---|---|---|---|
| **M 1** | Overall | 1.27 (1.05–1.48) | <0.001 | | 1.13 (1.10–1.15) | <0.001 | |
| | Gender | | | 0.002 | | | 0.003 |
| | Female | 1.74 (1.32–2.16) | <0.001 | | 1.17 (1.13–1.22) | <0.001 | |
| | Male | 0.94 (0.72–1.17) | <0.001 | | 1.10 (1.06–1.13) | <0.001 | |
| **M 2** | Overall | 1.00 (0.83–1.16) | <0.001 | | 1.17 (1.14–1.20) | <0.001 | |
| | Gender | | | 0.045 | | | <0.001 |
| | Female | 1.23 (0.94–1.52) | <0.001 | | 1.26 (1.21–1.31) | <0.001 | |
| | Male | 0.86 (0.66–1.06) | <0.001 | | 1.13 (1.09–1.16) | <0.001 | |
| **M 3** | Overall | 0.75 (0.62–0.88) | <0.001 | | 1.14 (1.11–1.17) | <0.001 | |
| | Gender | | | 0.018 | | | 0.002 |
| | Female | 1.02 (0.81–1.23) | <0.001 | | 1.22 (1.17–1.28) | <0.001 | |
| | Male | 0.59 (0.42–0.75) | <0.001 | | 1.10 (1.07–1.13) | <0.001 | |

**Fig 3. Multivariable regression analyses and sex-stratified analyses.** Multivariate regression analyses (**A**) and sex-stratified analyses (**B**) for the association of VAI with biological aging (KDMAge and KDMAgeAccel risk). Model 1 (M 1): unadjusted. Model 2 (M 2): demographic-adjusted (age, sex

[whole population only], race, education, status of mariage and poverty). Model 3 (M 3): adjusted for M 2 + lifestyle behaviors (smoking, alcohol consumption, M/VPA) and comorbidities (HTN, CVD, cancer, and CKD). VAI, visceral adiposity index; OR, odds ratio; KDMAge, Klemera-Doubal method age; KDMAgeAccel, KDMAge acceleration; CI, confidence interval.

Overall, VAI demonstrated significant positive associations with BA. Notably, females exhibited a 73% stronger association with KDMAge and 11% greater odds of KDMAgeAccel risk than males did per unit increase in VAI. The nonoverlapping CIs and significant interaction effects suggest that female visceral adiposity may have a significantly stronger association with BA than male visceral adiposity does.

## Nonlinear analysis

After fully adjusted covariates, RCSs revealed significant nonlinear relationships between VAI and BA across all cohorts (whole population, females, and males; $P_{nonlinear} < 0.001$). The dose-response curves demonstrated a biphasic upward trend, featuring a steep acceleration of BA followed by a sustained, gradual ascent as VAI increased (Fig 4). Threshold effect analyses (Table 2) revealed that the VAI thresholds were 3.41 for KDMAge and 2.54 for

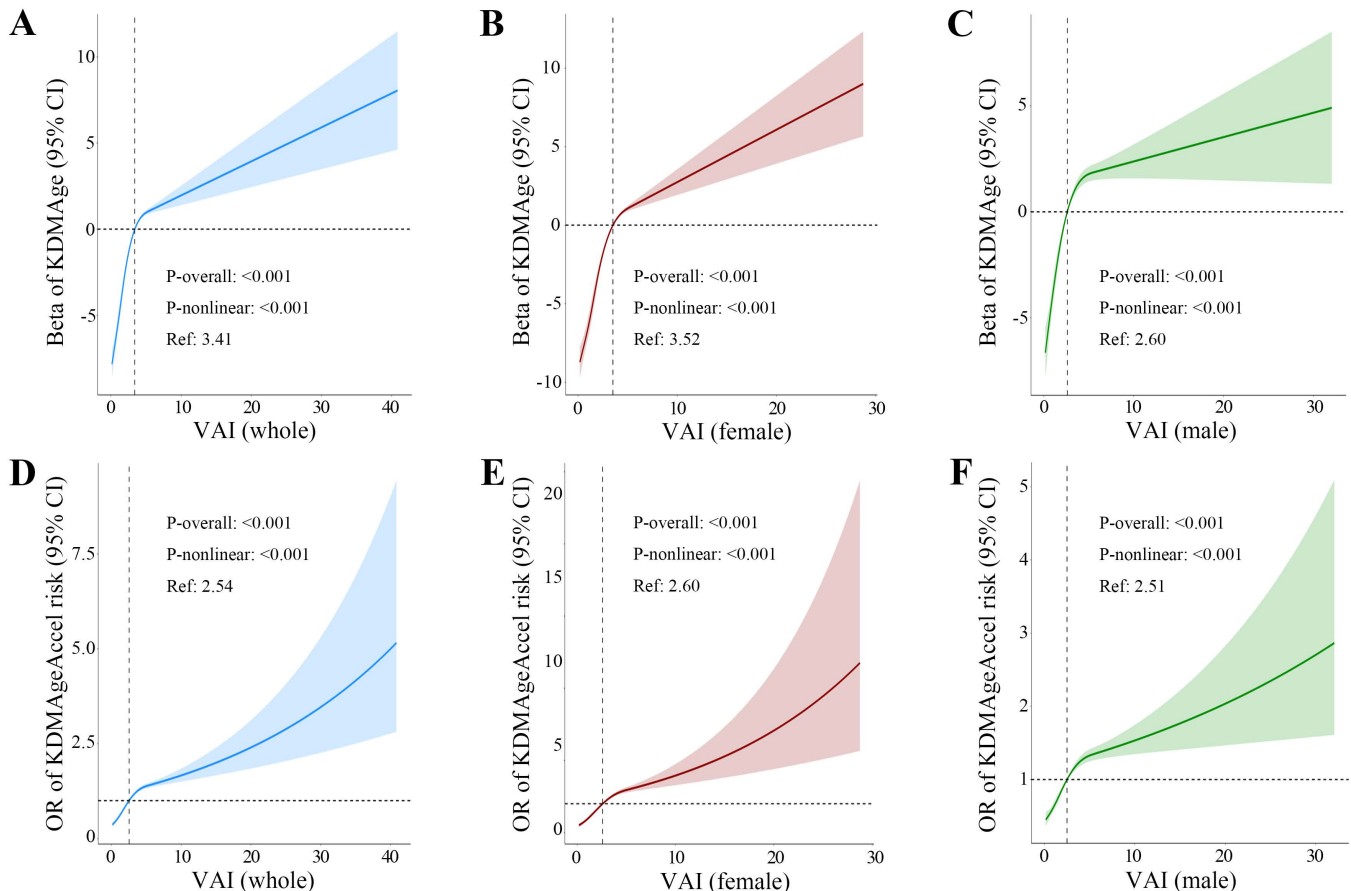

**Fig 4. Restricted cubic splines for the association of VAI with biological aging.** Associations between VAI and KDMAge/KDMAgeAccel risk among the whole population (A, D), females (B, E), and males (C, F). CI, confidence interval; OR, odds ratio; VAI, visceral adiposity index; KDMAge, Klemera-Doubal method age; KDMAgeAccel, KDMAge acceleration.

**Table 2. Threshold effect analyses of the association between VAI and biological aging.**

| | KDMAge | | KDMAgeAccel risk | |
|---|---|---|---|---|
| | β (95% CI) | *P*-value | OR (95% CI) | *P*-value |
| **Whole population** | | | | |
| Standard linear regression | 0.68 (0.60–0.75) | <0.001 | 1.11 (1.09–1.13) | <0.001 |
| Two-piecewise linear regression | | | | |
| K | 3.41 | | 2.54 | |
| < K | 2.17 (1.89–2.45) | <0.001 | 1.50 (1.41–1.61) | <0.001 |
| ≥ K | 0.20 (0.08–0.31) | <0.001 | 1.04 (1.02–1.06) | <0.001 |
| Likelihood ratio | | <0.001 | | <0.001 |
| **Females** | | | | |
| Standard linear regression | 0.87 (0.77–0.98) | <0.001 | 1.17 (1.14–1.20) | <0.001 |
| Two-piecewise linear regression | | | | |
| K | 3.52 | | 2.60 | |
| < K | 2.55 (2.22–2.89) | <0.001 | 1.75 (1.59–1.93) | <0.001 |
| ≥ K | 0.22 (0.04–0.39) | 0.014 | 1.04 (1.02–1.07) | 0.001 |
| Likelihood ratio | | <0.001 | | <0.001 |
| **Males** | | | | |
| Standard linear regression | 0.56 (0.45–0.67) | <0.001 | 1.08 (1.06–1.10) | <0.001 |
| Two-piecewise linear regression | | | | |
| K | 2.60 | | 2.51 | |
| < K | 2.46 (1.89–3.04) | <0.001 | 1.36 (1.24–1.50) | <0.001 |
| ≥ K | 0.23 (0.09–0.38) | 0.001 | 1.04 (1.02–1.06) | <0.001 |
| Likelihood ratio | | <0.001 | | <0.001 |

VAI, visceral adiposity index; KDMAge, Klemera-Doubal method age; KDMAgeAccel, KDMAge acceleration; CI, confidence interval; OR, odds ratio; K, inflection point.

KDMAgeAccel risk in the whole population. Below these thresholds, higher VAI is linked to substantially faster BA ($\beta_{KDMAge}$ = 2.17, 95% CI: 1.89–2.45; $OR_{KDMAgeAccel}$ = 1.50, 95% CI: 1.41–1.61). Above the thresholds, the associations diminished but remained significant ($\beta_{KDMAge}$ = 0.20, 95% CI: 0.08–0.31; $OR_{KDMAgeAccel}$ = 1.04, 95% CI: 1.02–1.06). Significant for all these associations ($P < 0.001$). Nonetheless, the threshold impact of the nonlinear relationships connecting VAI with BA varied between the male and female groups. Specifically, the inflection points (K) for the VAI–KDMAge nonlinear association was greater in females than in males ($K_{female}$ = 3.52 vs. $K_{male}$ = 2.60), though both pre-threshold effects and post-threshold effects showed comparable magnitudes across sexes. In contrast, the inflection points for VAI–KDMAgeAccel risk nonlinear associations were similar between sexes ($K_{female}$ = 2.60 vs. $K_{male}$ = 2.51), but differential threshold effects emerged: females exhibited significantly stronger pre-threshold effects ($OR_{female}$ = 1.75, 95% CI: 1.59–1.93 vs. $OR_{male}$ = 1.36, 95% CI: 1.24–1.50; $P < 0.001$), whereas post-threshold effects were comparable. The superior fit of nonlinear models over linear specifications was confirmed across all cohorts via likelihood ratio tests ($P_{likelihood\ ratio} < 0.001$).

These findings suggested that the nonlinear relationships between VAI and BA may differ between the sexes. Females may demonstrate a strong positive correlation with KDMAge across a wider VAI spectrum yet presented a greater risk of KDMAgeAccel even at VAI levels comparable to males. Conversely, males enter a period of slow increase of KDMAge at lower VAI thresholds while maintaining lower KDMAgeAccel risks at equivalent adiposity levels.

## Mediation analysis

**Association between VAI and HOMA-IR.** Using different covariate adjustment strategies in multivariable linear regression analyses, all models revealed a stable positive connection linking VAI to HOMA-IR. Table 3 indicates that within the completely adjusted model, a 1-unit higher VAI predicted a 0.42-unit elevation in HOMA-IR ($\beta = 0.42$, 95% CI: 0.32–0.52) across the whole population. Remarkably, despite the slightly elevated estimates observed for females in the sex-stratified analysis ($\beta_{female} = 0.50$, 95% CI: 0.33–0.66; $\beta_{male} = 0.36$, 95% CI: 0.24–0.48), the sex interaction did not attain statistical significance ($P_{interaction} = 0.308$). See S5 Table for details. The finding indicates that there might not be a discernible sex difference in the association between VAI and HOMA-IR. Stratifying VAI into quartiles revealed a clear dose-response relationship. Relative to Q1, Q4 participants showed significantly higher HOMA-IR ($\beta = 3.32$, 95% CI: 3.03–3.61; $P_{trend} < 0.001$). Similar relationships were observed for both sexes ($\beta_{female} = 3.13$, 95% CI: 2.82–3.45; $\beta_{male} = 3.43$, 95% CI: 2.99–3.86), with significant linear trends ($P_{trend} < 0.001$). All associations were statistically significant ($P < 0.001$). This finding suggests a strong positive association of visceral obesity with IR.

**Table 3. Association between VAI and HOMA-IR.**

|  | Whole population | | Females | | Males | |
|---|---|---|---|---|---|---|
|  | β (95% CI) | *P*-value | β (95% CI) | *P*-value | β (95% CI) | *P*-value |
| M 1 |  |  |  |  |  |  |
| VAI continue | 0.47 (0.36–0.57) | <0.001 | 0.54 (0.36–0.72) | <0.001 | 0.41 (0.29–0.54) | <0.001 |
| VAI quantile |  |  |  |  |  |  |
| Q1 | 0.00 (Reference) |  | 0.00 (Reference) |  | 0.00 (Reference) |  |
| Q2 | 0.74 (0.57–0.91) | <0.001 | 0.83 (0.60–1.06) | <0.001 | 0.64 (0.40–0.87) | <0.001 |
| Q3 | 1.83 (1.61–2.05) | <0.001 | 1.61 (1.46–1.77) | <0.001 | 2.11 (1.70–2.51) | <0.001 |
| Q4 | 3.69 (3.38–3.99) | <0.001 | 3.45 (3.11–3.80) | <0.001 | 3.88 (3.44–4.32) | <0.001 |
| *P*-trend |  | <0.001 |  | <0.001 |  | <0.001 |
| M 2 |  |  |  |  |  |  |
| VAI continue | 0.46 (0.35–0.56) | <0.001 | 0.53 (0.35–0.70) | <0.001 | 0.40 (0.28–0.53) | <0.001 |
| VAI quantile |  |  |  |  |  |  |
| Q1 | 0.00 (Reference) |  | 0.00 (Reference) |  | 0.00 (Reference) |  |
| Q2 | 0.73 (0.56–0.91) | <0.001 | 0.80 (0.59–1.02) | <0.001 | 0.60 (0.35–0.85) | <0.001 |
| Q3 | 1.80 (1.57–2.03) | <0.001 | 1.55 (1.38–1.71) | <0.001 | 2.06 (1.64–2.47) | <0.001 |
| Q4 | 3.62 (3.32–3.92) | <0.001 | 3.37 (3.04–3.71) | <0.001 | 3.82 (3.37–4.26) | <0.001 |
| *P*-trend |  | <0.001 |  | <0.001 |  | <0.001 |
| M 3 |  |  |  |  |  |  |
| VAI continue | 0.42 (0.32–0.52) | <0.001 | 0.50 (0.33–0.66) | <0.001 | 0.36 (0.24–0.48) | <0.001 |
| VAI quantile |  |  |  |  |  |  |
| Q1 | 0.00 (Reference) |  | 0.00 (Reference) |  | 0.00 (Reference) |  |
| Q2 | 0.67 (0.49–0.84) | <0.001 | 0.76 (0.56–0.95) | <0.001 | 0.50 (0.25–0.76) | <0.001 |
| Q3 | 1.62 (1.38–1.86) | <0.001 | 1.42 (1.24–1.59) | <0.001 | 1.82 (1.38–2.26) | <0.001 |
| Q4 | 3.32 (3.03–3.61) | <0.001 | 3.13 (2.82–3.45) | <0.001 | 3.43 (2.99–3.86) | <0.001 |
| *P*-trend |  | <0.001 |  | <0.001 |  | <0.001 |

Model 1 (M 1): unadjusted

Model 2 (M 2): demographic-adjusted (age, sex [whole population only], race, education, status of mariage and poverty).

Model 3 (M 3): adjusted for M 2 + lifestyle behaviors (smoking, alcohol consumption, M/VPA) and comorbidities (HTN, CVD, cancer, and CKD).

VAI, visceral adiposity index; CI, confidence interval; HOMA-IR, homeostasis model assessment of insulin resistance.

## Association between HOMA-IR and biological aging

HOMA-IR was consistently and markedly positively correlated with BA (KDMAge and KDMAgeAccel risk) across all the models. Table 4 showed all the models for the relationship between HOMA-IR and KDMAge. Full adjustment revealed a 0.35-year KDMAge acceleration per HOMA-IR unit increase (β = 0.35, 95% CI: 0.27–0.43) in the whole population, with comparable effects in females (β = 0.35, 95% CI: 0.25–0.44) and males (β = 0.34, 95% CI: 0.23–0.45). When HOMA-IR was divided into quartiles, the top quartile showed a stronger relationship with KDMAge than did the lowest quartile (β = 7.01, 95% CI: 6.28–7.73). Significant associations were observed in both sexes ($β_{female}$ = 6.77, 95% CI: 6.03–7.51; $β_{male}$ = 7.84, 95% CI: 6.74–8.93), with significant linear trends ($P_{trend}$ < 0.001). Every tested association demonstrated robust significance (P < 0.001).

Table 5 showed all the models for the relationship between HOMA-IR and KDMAgeAccel risk. In the completely adjusted models of the whole population and sex-specific cohorts, a one-unit rise in HOMA-IR was linked to a 7% increase in the risk of KDMAgeAccel, and elevated quartiles of HOMA-IR were related to greater risks than Q1 across all cohorts (P < 0.001). No significant sex interaction was found for HOMA-IR and BA ($P_{interaction}$ > 0.05). See S5 Table for

**Table 4. Association between HOMA-IR and KDMAge.**

|  | Whole population | | Females | | Males | |
|---|---|---|---|---|---|---|
|  | β (95% CI) | *P*-value | β (95% CI) | *P*-value | β (95% CI) | *P*-value |
| M 1 |  |  |  |  |  |  |
| HOMA-IR continue | 0.72 (0.59–0.86) | <0.001 | 0.82 (0.68–0.97) | <0.001 | 0.69 (0.50–0.89) | <0.001 |
| HOMA-IR quantile |  |  |  |  |  |  |
| Q1 | 0.00 (Reference) |  | 0.00 (Reference) |  | 0.00 (Reference) |  |
| Q2 | 3.45 (2.58–4.33) | <0.001 | 4.01 (2.96–5.05) | <0.001 | 3.45 (2.25–4.65) | <0.001 |
| Q3 | 7.41 (6.45–8.37) | <0.001 | 7.26 (6.04–8.49) | <0.001 | 7.82 (6.59–9.04) | <0.001 |
| Q4 | 13.30 (12.43–14.17) | <0.001 | 12.71 (11.65–13.76) | <0.001 | 15.05 (13.78–16.32) | <0.001 |
| *P*-trend |  | <0.001 |  | <0.001 |  | <0.001 |
| M 2 |  |  |  |  |  |  |
| HOMA-IR continue | 0.53 (0.42–0.63) | <0.001 | 0.53 (0.41–0.66) | <0.001 | 0.53 (0.38–0.67) | <0.001 |
| HOMA-IR quantile |  |  |  |  |  |  |
| Q1 | 0.00 (Reference) |  | 0.00 (Reference) |  | 0.00 (Reference) |  |
| Q2 | 2.72 (1.98–3.47) | <0.001 | 3.35 (2.47–4.22) | <0.001 | 2.60 (1.54–3.67) | <0.001 |
| Q3 | 5.43 (4.67–6.19) | <0.001 | 5.77 (4.87–6.68) | <0.001 | 5.24 (4.21–6.26) | <0.001 |
| Q4 | 9.82 (9.10–10.55) | <0.001 | 9.20 (8.36–10.04) | <0.001 | 11.26 (10.20–12.33) | <0.001 |
| *P*-trend |  | <0.001 |  | <0.001 |  | <0.001 |
| M 3 |  |  |  |  |  |  |
| HOMA-IR continue | 0.35 (0.27–0.43) | <0.001 | 0.35 (0.25–0.44) | <0.001 | 0.34 (0.23–0.45) | <0.001 |
| HOMA-IR quantile |  |  |  |  |  |  |
| Q1 | 0.00 (Reference) |  | 0.00 (Reference) |  | 0.00 (Reference) |  |
| Q2 | 2.43 (1.71–3.14) | <0.001 | 2.95 (2.15–3.75) | <0.001 | 2.20 (1.15–3.25) | <0.001 |
| Q3 | 4.10 (3.28–4.93) | <0.001 | 4.70 (3.82–5.58) | <0.001 | 3.86 (2.77–4.96) | <0.001 |
| Q4 | 7.01 (6.28–7.73) | <0.001 | 6.77 (6.03–7.51) | <0.001 | 7.84 (6.74–8.93) | <0.001 |
| *P*-trend |  | <0.001 |  | <0.001 |  | <0.001 |

Model 1 (M 1): unadjusted

Model 2 (M 2): demographic-adjusted (age, sex [whole population only], race, education, status of mariage and poverty).

Model 3 (M 3): adjusted for M 2 + lifestyle behaviors (smoking, alcohol consumption, M/VPA) and comorbidities (HTN, CVD, cancer, and CKD).

HOMA-IR, homeostasis model assessment of insulin resistance; CI, confidence interval; KDMAge, Klemera-Doubal method age.

**Table 5. Association between HOMA-IR and KDMAgeAccel risk.**

| | Whole population | | Females | | Males | |
|---|---|---|---|---|---|---|
| | OR (95% CI) | *P*-value | OR (95% CI) | *P*-value | OR (95% CI) | *P*-value |
| M 1 | | | | | | |
| HOMA-IR continue | 1.08 (1.06–1.10) | <0.001 | 1.08 (1.05–1.11) | <0.001 | 1.08 (1.05–1.11) | <0.001 |
| HOMA-IR quantile | | | | | | |
| Q1 | 1.00 (Reference) | | 1.00 (Reference) | | 1.00 (Reference) | |
| Q2 | 1.45 (1.26–1.67) | <0.001 | 1.57 (1.29–1.91) | <0.001 | 1.30 (1.06–1.59) | 0.013 |
| Q3 | 1.84 (1.59–2.12) | <0.001 | 2.23 (1.85–2.68) | <0.001 | 1.63 (1.37–1.93) | <0.001 |
| Q4 | 3.04 (2.68–3.45) | <0.001 | 3.19 (2.71–3.74) | <0.001 | 3.06 (2.61–3.58) | <0.001 |
| *P*-trend | | <0.001 | | <0.001 | | <0.001 |
| M 2 | | | | | | |
| HOMA-IR continue | 1.10 (1.08–1.12) | <0.001 | 1.10 (1.07–1.13) | <0.001 | 1.10 (1.07–1.13) | <0.001 |
| HOMA-IR quantile | | | | | | |
| Q1 | 1.00 (Reference) | | 1.00 (Reference) | | 1.00 (Reference) | |
| Q2 | 1.52 (1.32–1.76) | <0.001 | 1.62 (1.32–1.98) | <0.001 | 1.42 (1.16–1.74) | <0.001 |
| Q3 | 2.15 (1.85–2.50) | <0.001 | 2.49 (2.03–3.05) | <0.001 | 2.03 (1.69–2.43) | <0.001 |
| Q4 | 4.01 (3.54–4.55) | <0.001 | 4.11 (3.45–4.88) | <0.001 | 4.18 (3.58–4.88) | <0.001 |
| *P*-trend | | <0.001 | | <0.001 | | <0.001 |
| M 3 | | | | | | |
| HOMA-IR continue | 1.07 (1.05–1.09) | <0.001 | 1.07 (1.05–1.10) | <0.001 | 1.07 (1.05–1.10) | <0.001 |
| HOMA-IR quantile | | | | | | |
| Q1 | 1.00 (Reference) | | 1.00 (Reference) | | 1.00 (Reference) | |
| Q2 | 1.53 (1.32–1.77) | <0.001 | 1.61 (1.31–1.97) | <0.001 | 1.44 (1.17–1.78) | <0.001 |
| Q3 | 1.96 (1.67–2.30) | <0.001 | 2.33 (1.90–2.86) | <0.001 | 1.87 (1.53–2.28) | <0.001 |
| Q4 | 3.28 (2.87–3.76) | <0.001 | 3.43 (2.87–4.09) | <0.001 | 3.41 (2.88–4.04) | <0.001 |
| *P*-trend | | <0.001 | | <0.001 | | <0.001 |

Model 1 (M 1): unadjusted

Model 2 (M 2): demographic-adjusted (age, sex [whole population only], race, education, status of mariage and poverty).

Model 3 (M 3): adjusted for M 2 + lifestyle behaviors (smoking, alcohol consumption, M/VPA) and comorbidities (HTN, CVD, cancer, and CKD).

HOMA-IR, homeostasis model assessment of insulin resistance; KDMAge, Klemera-Doubal method age; OR, odds ratio; CI, confidence interval.

details. The results underscored a strong positive association of IR with BA, with consistent patterns across linear and categorical analyses.

## Mediation effect of HOMA-IR on the association between VAI and biological aging

After fully adjusted, results revealed an indirect effect of VAI on BA through HOMA-IR in both the whole population and sex-stratified subgroups (Fig 5). In the whole population, HOMA-IR mediated 18.92% (β = 0.145, 95% CI: 0.091–0.159) and 20.90% (β = 0.005, 95% CI: 0.003–0.006) of VAI's effects on KDMAge and KDMAgeAccel risk, respectively (see S6 Table for details). Sex-stratified analyses revealed notable differences in the mediation proportions of HOMA-IR in mediating the relationship of VAI and BA. In females, HOMA-IR mediated 12.71% (VAI–KDMAge: β = 0.139, 95% CI: 0.077–0.164) and 10.39% (VAI–KDMAgeAccel risk: β = 0.004, 95% CI: 0.002–0.006). Conversely, males showed higher mediation: 21.67% (VAI–KDMAge: β = 0.125, 95% CI: 0.071–0.161) and 27.09% (VAI–KDMAgeAccel risk: β = 0.005, 95% CI: 0.002–0.006). All these indirect effects were statistically significant (*P* < 0.001).

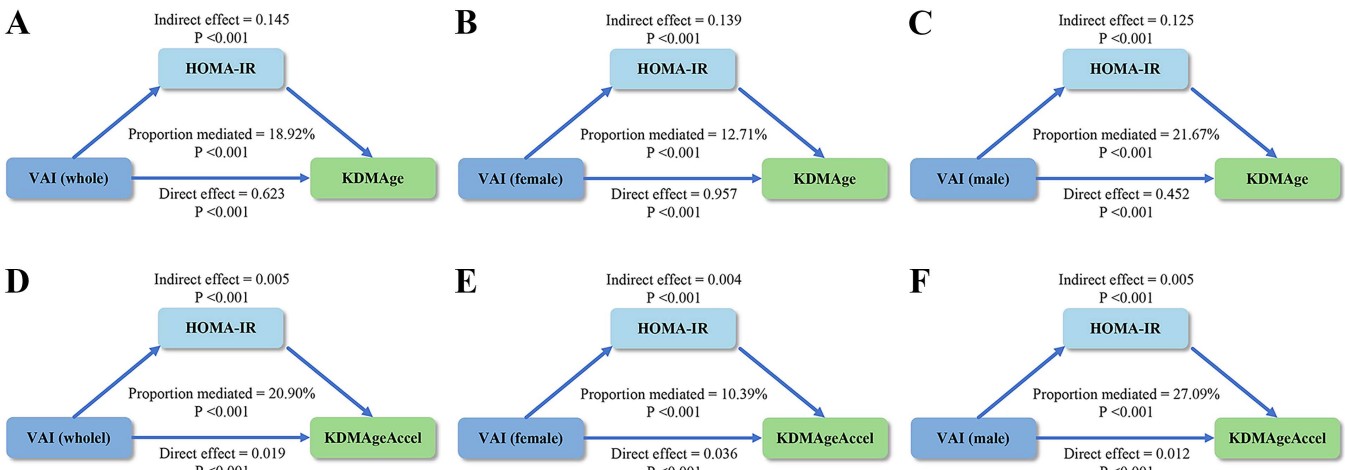

**Fig 5. Mediation effects between VAI and biological aging.** The Mediation effects of HOMA-IR on the associations between VAI and KDMAge/KDMAgeAccel risk among the whole population (A, D), females (B, E), and males (C, F). VAI, visceral adiposity index; KDMAge, Klemera-Doubal method age; HOMA-IR, homeostasis model assessment of insulin resistance.

Notably, the absolute estimates of the indirect effect through HOMA-IR were comparable between sexes, given the overlapping CIs. The sex disparity in the mediation proportions may stem from divergent direct effects. Compared with females, male participants displayed a significantly attenuated direct effect of VAI on KDMAge ($\beta_{male}$ = 0.452 vs. $\beta_{female}$ = 0.957) and KDMAgeAccel risk ($\beta_{male}$ = 0.012 vs. $\beta_{female}$ = 0.036), with nonoverlapping CIs confirming sex-specific effect magnitudes. This pattern persisted after adjusting for HOMA-IR, as evidenced by significant sex interaction terms ($P_{interaction}$ < 0.05; see S7 Table). Therefore, although the indirect effects of HOMA-IR on the VAI–BA association were comparable between sexes, the diminished direct effect in males contributed to a lower total effect, ultimately yielding a higher mediation proportion of HOMA-IR in males compared to females. Current evidence suggests that IR may be a potential key mechanistic mediator connecting visceral adiposity to BA in males, who may demonstrate greater reliance on IR-associated pathways than females. However, longitudinal studies are necessary to establish causal mediation mechanisms across different sexes, particularly given the inherent limitations of cross-sectional data in delineating mechanistic hierarchies.

### Sensitivity analyses

**Additional adjustment for DM and HDL.** VAI remained robustly positively associated with BA in the whole population ($\beta_{KDMAge}$ = 0.71; $OR_{KDMAgeAccel}$ = 1.12; $P$ < 0.001), with significant sex interactions ($P_{interaction}$ < 0.05) after additional adjustment for DM and HDL. Later analysis demonstrated that female participants exhibited a considerably stronger association than their male counterparts ($\beta_{KDMAge}$ = 1.15 vs. 0.42; $OR_{KDMAgeAccel}$ = 1.25 vs. 1.07; $P$ < 0.001). Additionally, females in the highest VAI quartile exhibited steeper dose-response slopes compared to males within the same quartile ($\beta_{KDMAge}$ = 12.30 vs. 6.28; $OR_{KDMAgeAccel}$ = 5.70 vs. 2.16), with significant linear trends ($P_{trend}$ < 0.001). Detailed information can be found in S8 and S9 Tables.

The re-evaluated RCS curves (S1 Fig) and threshold analyses (S10 Table) revealed similar results in VAI–BA relationships across all cohorts. Specifically, females demonstrated lightly higher VAI threshold point (K) for KDMAge ($K_{female}$ = 2.85 vs. $K_{male}$ = 2.53) with significantly steeper pre-threshold slopes ($\beta_{female}$ = 5.39 vs. $\beta_{male}$ = 2.73; $P$ < 0.001), though post-threshold effects converged. Although VAI–KDMAgeAccel risk nonlinear associations showed comparable inflection points ($K_{female}$ = 2.29 vs. $K_{male}$ = 2.50), females presented significantly greater pre-threshold risk amplification ($OR_{female}$ = 2.67

vs. $OR_{male} = 1.35$; $P < 0.001$), with comparable post-threshold effects. These results further validate the persistence of sex differences in both linear and nonlinear associations.

Mediation analysis confirmed a significant indirect effect of HOMA-IR on the VAI–BA association across all cohorts, after additional adjustment for DM and HDL (S11 Table). In the whole population, HOMA-IR mediated 8.24% of VAI's total effect on KDMAge and 10.62% of its effect on KDMAgeAccel risk. Sex-specific analyses revealed notably lower mediation proportions in females: 5.80% for KDMAge and 5.42% for KDMAgeAccel risk. In contrast, males exhibited substantially higher proportions: 10.54% and 14.46%, respectively. All these indirect effects were statistically significant ($P < 0.001$).

**Exclusion of participants with DM.** To evaluate the potential confounding effect of overt diabetes, sensitivity analyses were performed after excluding all individuals with diagnosed DM (n = 3708, 19.03%). VAI maintained robust positive associations with BA in the whole population ($\beta_{KDMAge} = 0.77$, 95% CI: 0.62–0.93; $OR_{KDMAgeAccel} = 1.15$, 95% CI: 1.12–1.18; $P < 0.001$), with significant sex interaction effects ($P_{interaction} < 0.01$). Sex-stratified analyses revealed significantly stronger associations in females than males for both continuous VAI (KDMAge: $\beta_{female} = 1.13$, 95% CI: 0.81–1.46 vs. $\beta_{male} = 0.59$, 95% CI: 0.43–0.75; KDMAgeAccel: $OR_{female} = 1.27$, 95% CI: 1.21–1.34 vs. $OR_{male} = 1.10$, 95% CI: 1.06–1.14; $P < 0.001$) and highest VAI quartile comparisons (KDMAge: $\beta_{female} = 6.97$, 95% CI: 5.97–7.96 vs. $\beta_{male} = 5.97$, 95% CI: 4.77–7.17; KDMAgeAccel: $OR_{female} = 3.46$, 95% CI: 2.80–4.27 vs. $OR_{male} = 2.43$, 95% CI: 1.96–3.02; $P < 0.001$). Significant linear trends persisted across VAI quartiles in both sexes ($P_{trend} < 0.001$). Complete results are presented in S12 and S13 Tables.

Re-evaluated RCSs (S2 Fig) and threshold analyses (S14 Table) confirmed the persistence of sex-specific patterns in VAI–BA relationships. Females demonstrated stronger associations with KDMAge across a broader VAI range (inflection point: $K_{female} = 3.41$ vs. $K_{male} = 2.51$), despite comparable effect magnitudes both below ($\beta_{female} = 2.48$, 95% CI: 2.12–2.84 vs. $\beta_{male} = 2.35$, 95% CI: 1.73–2.96; $P < 0.001$) and above thresholds ($\beta_{female} = 0.18$, 95% CI: 0.02–0.39 vs. $\beta_{male} = 0.24$, 95% CI: 0.08–0.39; $P < 0.01$). Similarly, for KDMAgeAccel risk, females exhibited markedly lower threshold inflection points ($K_{female} = 0.60$ vs. $K_{Kmale} = 2.53$) with stronger associations above the threshold ($OR_{female} = 1.18$, 95% CI: 1.14–1.22 vs. $OR_{male} = 1.04$, 95% CI: 1.02–1.07; $P < 0.01$).

Mediation analyses (S15 Table) revealed consistent sex differences in HOMA-IR mediation proportions after excluding diabetic participants. HOMA-IR mediated 17.85% of the VAI–KDMAge association and 22.59% of the VAI–KDMAgeAccel risk association in the whole population. Mediation proportions remained significantly higher in males than females for both KDMAge (21.56% vs. 11.11%) and KDMAgeAccel risk (29.46% vs. 18.98%). All indirect effects remained statistically significant ($P < 0.001$). Sensitivity analyses validated the robustness of the findings presented in this study.

## Subgroup analyses

Consistent with our primary findings, stratified analyses across racial/ethnic subgroups (Mexican American, Non-Hispanic White, Non-Hispanic Black, Other) revealed persistently positive associations between VAI, BA, and IR in all populations. Three key patterns emerged: (a) VAI–HOMA-IR associations remained significantly positive across all racial/ethnic groups in the whole cohort and sex-stratified subgroups (S16 Table). Formal interaction testing indicated significant effect modification by race/ethnicity in the whole population ($P_{interaction} = 0.044$) and among females ($P_{interaction} = 0.018$). (b) HOMA-IR–BA relationships demonstrated consistent positive effects in all racial/ethnic subgroups without significant interaction ($P_{interaction} > 0.05$ for whole, male, and female cohorts; S17 Table), indicating race-invariant mediation pathways. (c) The VAI–BA associations maintained significant positive effects across racial/ethnic strata (S18 Table), with significant interaction effects observed in the whole population (KDMAge: $P_{interaction} = 0.021$; KDMAgeAccel: $P_{interaction} = 0.006$) and among males (KDMAge: $P_{interaction} = 0.025$; KDMAgeAccel: $P_{interaction} = 0.026$). These findings confirm the reliability of our primary observations while identifying potential racial/ethnic differences, which enhances the generalizability of the study results and provides a basis for targeted interventions.

A subsequent age-stratified analysis of all subgroups (20–39 years, 40–59 years, ≥ 60 years) demonstrated a significant positive association between VAI and BA in all three cohorts ($P < 0.05$ for all age groups; S19 Table). This finding is

aligned with the conclusions drawn from our preliminary analysis. Formal interaction testing revealed significant effect modification by age in the whole population for both KDMAge ($P_{interaction}$ = 0.017) and KDMAgeAccel risk ($P_{interaction}$ = 0.039). Notably, the strongest associations emerged in young adults (20−39 years), where each unit VAI increase corresponded to 1.05-year KDMAge (β = 1.05, 95% CI: 0.83–1.28; P < 0.001) and 22% higher KDMAgeAccel risk (OR = 1.22, 95% CI: 1.16–1.29; P < 0.001). Surprisingly, sex-stratified analyses showed no significant age interaction in either male or female subgroups ($P_{interaction}$ > 0.05). These findings reveal a previously unrecognized vulnerability of younger adults to visceral adiposity-induced BA, highlighting critical windows for preventive intervention.

## Discussion

Based on NHANES 1999–2018, this national study of 19486 (9732 females and 9754 males) American adult participants revealed significant positive associations between VAI and BA (KDMAge and KDMAgeAccel risk). Nonlinear analyses revealed a biphasic pattern: a steep acceleration of BA followed by a sustained, gradual increase as VAI increased. Significant sex differences were observed, with females demonstrating a 73% stronger VAI–KDMAge association and an 11% higher VAI–KDMAgeAccel risk compared to males. Threshold analyses further indicated that females demonstrated a strong positive correlation with KDMAge across a wider VAI spectrum (< 3.52 vs. < 2.60) and presented a greater risk of KDMAgeAccel (75% vs. 36%) even at VAI levels comparable to males. Mediation analysis revealed that HOMA-IR significantly mediates the VAI–BA link. Despite similar effect values in both sexes, HOMA-IR contributed more to males' VAI–KDMAge (21.67% vs. 12.71%) and VAI–KDMAgeAccel risk (27.09% vs. 10.39%) compared to females. In the sensitivity analysis, our findings maintained their statistical significance following adjustments for additional variables. This study represents, as far as we know, the first investigation into how HOMA-IR mediates the associations of VAI with BA (KDMAge and KDMAgeAccel risk) across sexes. These findings elucidate sex-specific association between visceral adiposity and BA, with IR demonstrating differential mediation patterns between sexes. This study may provide potential guidance for the future development of sex-specific public health strategy of obesity and aging.

Our findings, align with prior evidence, demonstrate that VAI may be a robust biomarker for BA. CVD is a well-documented driver of accelerated aging mechanisms [28]. Further research has shown a nonlinear relationship between VAI and cardiovascular mortality. Specifically, VAI levels below 2.49 are followed by a 122% higher risk of cardiovascular death per 1-unit increase, potentially with a greater effect observed in females [10]. IR, another potential contributor to BA, is strongly and positively correlated with VAI levels, exhibiting a 28% escalation in IR risk per unit increment in VAI [29,30]. It is noteworthy that sedentary behavior is associated with an elevated VAI, whereas physical inactivity itself accelerates the aging process [31,32]. VAI emerges from these investigations as a potential modulator of aging, with underlying mechanisms that may originate from vascular dysfunction, IR, chronic inflammation, and lifestyle factors.

A study on cardiovascular metabolism revealed a key mechanism by which visceral adiposity drives aging [15]. Visceral adipose tissue discharges abundant free fatty acids into the bloodstream, which accumulate in vascular endothelial cells, impairing endothelial function and accelerating atherosclerosis and cardiovascular events, ultimately reducing cardiac output and promoting systemic ischemia-hypoxia to drive aging. Furthermore, chronic inflammation caused by visceral adipose tissue can accelerate organismal aging via two main mechanisms. On the one hand, inflammation directly harms cells by triggering apoptosis and hindering repair, and can also cause oxidative stress, metabolic disturbances, and organ dysfunction [33]. On the other hand, visceral fat-derived cytokines also disrupt gut microbiota homeostasis and intestinal barrier integrity, exacerbating systemic inflammation and immune dysfunction to accelerate aging. Those demonstrating greater VAI values frequently exhibit inactive lifestyles and smoking patterns, reflecting unhealthy habits that engage with biological mechanisms to speed up the aging process [10,34].

IR is a pathological condition in which the biological response of the body to insulin is diminished. The IR derived from visceral adiposity is one of the central mechanisms linking VAI to BA, driven by proinflammatory cytokines, mitochondrial dysfunction, lipotoxicity, endoplasmic reticulum stress, as well as epigenetic dysregulation. Visceral adipose tissue

expansion triggers systemic low-grade inflammation via macrophage infiltration and the secretion of proinflammatory cytokines, which activate JNK/IKKβ kinases to suppress insulin receptor substrate phosphorylation and impair the PI3K/Akt signaling pathway, thereby disrupting glucose uptake and promoting IR [35,36]. Concurrently, lipotoxicity, characterized by the accumulation of ceramide and diacylglycerol, induces mitochondrial fragmentation and oxidative stress, worsening aberrant fat deposition in muscle and liver tissues, which further disrupts insulin-mediated glucose metabolism [37,38]. These processes are compounded by endoplasmic reticulum stress, which triggers the unfolded protein response, impairing insulin synthesis in pancreatic β-cells and promoting adipokine imbalance (such as reduced adiponectin and increased resistin) [39,40]. Genetic polymorphisms and epigenetic dysregulation further exacerbate IR by altering insulin signaling and metabolic homeostasis [41].

We found via mediation analysis that a higher VAI was correlated with higher HOMA-IR, whereas HOMA-IR showed a consistent positive relationship with BA. The elevation of HOMA-IR, a well-established indicator of insulin resistance, not only reflects diminished whole-body insulin sensitivity but also exhibits significant associations with oxidative stress, chronic inflammation, and metabolic toxicity. These pathological processes collectively promote genomic instability through DNA damage, telomere shortening, and p53-dependent senescent cell accumulation [42]. Recent studies emphasize that IR-induced mitochondrial reactive oxygen species overproduction and heterochromatin erosion activate innate immune pathways, thereby accelerating neuronal aging and cognitive decline, a mechanism also observed in primate models of brain aging [43]. Collectively, these findings underscore the potential for VAI to serve as a therapeutic target for mitigating aging. Nevertheless, this proposition merits empirical substantiation in prospective longitudinal designs.

Our data revealed that females exhibited significantly stronger VAI-related aging associations than their male counterparts. A study revealed that obese females presented shorter leukocyte telomere lengths compared to obese males [44]. This finding suggested that females may be more prone to obesity-induced cellular damage, which contributes to accelerated aging. HOMA-IR exhibited analogous mediator effect values across sexes. However, the proportion of HOMA-IR contributing to the link between VAI and aging was diminished among females, which suggests that females may be more dependent on non-IR mediators. The potential mechanisms underlying this phenomenon include sex-specific estrogen dynamics, cardiovascular inflammation sensitivity, comorbidities, and depression. Importantly, fluctuations in female-specific estrogen, driven by menstrual cycles, pregnancy, lactation, and menopausal transitions, profoundly alter adipose metabolism and endocrine regulation [45]. Decreased estrogen levels attenuate ERα-mediated protective signaling, triggering adipose redistribution from subcutaneous (e.g., hip/thigh) to visceral depots [46] while increasing lipoprotein lipase activity in adipocytes to increase visceral fat accumulation [47]. Visceral adipose overexpression of aromatase further promotes local androgen-to-estrogen conversion, exacerbating visceral adiposity [48]. These processes collectively drive visceral fat expansion and dysfunction, which serve as key sources of proinflammatory cytokines and senescence-associated secretory phenotypes that accelerate systemic aging [49]. According to research findings, women demonstrate augmented cardiovascular sensitivity to inflammation. This heightened sensitivity is characterized by the propensity of visceral fat-driven inflammatory responses to induce more pronounced endothelial dysfunction than in men [50]. Notably, elevated VAI levels in females are strongly associated with increased carotid atherosclerosis risk, a correlation that is absent in males [51]. Midlife-to-later-life obesity metrics in women exhibit a stronger correlation with aging-related comorbidities, including CVD, DM, and HTN [52]. This finding suggests a heightened cumulative burden of metabolic dysfunction. Additionally, both the psychological stress and the heightened inflammatory cytokine levels associated with obesity may serve to intensify depressive symptoms in females [53], which in turn accelerate cellular aging via reactive oxygen species-mediated DNA damage and senescence pathways [54]. While the underlying sex-specific mechanisms by which VAI influences aging have not been thoroughly understood, this evidence stresses the need for prospective cohort studies to verify causal associations to further elucidate the effects of visceral adiposity on aging.

This investigation possesses multiple strengths. First, we are the first to uncover the previously unrecognized relationship binding VAI with BA (KDMAge and KDMAgeAccel risk) while identifying the mediating role of HOMA-IR in these

relationships and revealing pronounced sex-specific disparities. Second, our utilization of NHANES' probability sampling design satisfies the statistical requirements for nationwide inference. Third, the robustness of these associations was rigorously validated through multivariable-adjusted models and sensitivity analysis. Our cross-sectional results suggest that VAI could function as a sex-stratified biomarker of BA status, providing hypotheses for future longitudinal studies to validate its predictive ability for aging trajectories and to explore sex-specific prevention methods.

Nevertheless, the study has four limitations. First, the primary limitation of this study stems from its cross-sectional design, which inherently precludes definitive causal inference regarding the temporal sequence between VAI and BA, particularly in elucidating sex-specific mechanisms. While our analyses revealed significant associations and mediation effects, this design cannot establish whether elevated VAI precedes and contributes to accelerated BA, or conversely, whether accelerated BA itself drives increases in VAI. Compelling evidence suggests that age-related lipid turnover decline and adipose tissue dysfunction may constitute reverse causal pathways. Specifically, longitudinal data demonstrate that adipose lipid turnover decreases by approximately 4.7% annually with aging, disrupting the dynamic equilibrium of lipid storage and clearance, even under constant caloric intake, potentially driving weight gain exceeding 20% over decades [55]. This dysregulation is particularly pronounced in visceral adipose tissue due to diminished preadipocyte differentiation capacity, reduced lipolytic enzyme activity (e.g., hormone-sensitive lipase), and blunted catecholamine responsiveness, collectively impairing lipid mobilization. Notably, the recent identification of CP-A adipose progenitor cells, a subpopulation emerging during middle age that exhibits hyperactive adipogenic potential through LIFR-STAT3 signaling activation, may further explain age-dependent visceral adipose tissue expansion [56]. Concurrently, adipose tissue senescence is characterized by chronic low-grade inflammation involving increased immune cell infiltration (e.g., TNF-α, IL-6), tissue fibrosis, and adipokine dysregulation (e.g., reduced adiponectin, elevated leptin), establishing a self-perpetuating vicious cycle whereby aging promotes adipose dysfunction that subsequently exacerbates systemic metabolic deterioration and accelerates organismal decline [57]. Additional contributors to age-associated visceral adiposity include muscle mass loss, progressive reductions in physical activity levels, and hormonal alterations [11,58,59]. Critically, the observed mediation proportion of HOMA-IR, while statistically significant, must also be interpreted cautiously within this bidirectional framework, as aging-related metabolic dysfunction may simultaneously elevate HOMA-IR independently of prior adiposity changes [60]. Importantly, this potential reverse causation may influence our key finding of stronger VAI–BA associations in females. Estrogen, predominantly in premenopausal women, maintains subcutaneous fat distribution and functionality while delaying visceral adipose tissue accumulation through ERα-mediated mechanisms; however, the abrupt estrogen decline during menopause may precipitate rapid visceral adipogenesis, potentially amplifying the observed VAI–BA relationship in female cohorts [59]. Furthermore, the biphasic nonlinear associations and threshold effects we identified may reflect shifting dominance between causal directions: below VAI thresholds, traditional "adiposity-driven aging" mechanisms (lipotoxicity, inflammation) likely predominate, whereas above thresholds, aging-induced metabolic dysregulation may progressively disrupt adipose homeostasis, creating a self-reinforcing pathological loop [57]. Consequently, the strength and sex-specific patterns of the associations reported here require validation in longitudinal studies tracking temporal dynamics of VAI, HOMA-IR, and BA biomarkers to establish precedence. Future experimental interventions targeting visceral adiposity reduction would provide stronger evidence for causal pathways and their sexual dimorphism. Second, despite the rigorous adjustment for numerous confounders, residual biases from other variables (e.g., genetic/epigenetic predispositions, dietary components, working conditions, or psychosocial conditions) may persist. Consequently, future studies integrating multi-omics data to refine confounding control may be necessary. Third, VAI lacks universally standardized clinical thresholds. Although we identified aging-related inflection points, further validation is needed to ascertain their generalizability and clinical utility across different populations. Fourth, while the biological age of the Klemera-Doubal method captures multidimensional aging features, incorporating complementary biomarkers (e.g., homeostatic dysregulation, allostatic load, and phenotypic age) in future studies could enhance mechanistic insights into visceral adiposity–aging interplay [8].

## Conclusions

Our findings demonstrate a significant positive association between VAI and BA, with a stronger correlation observed in females and a greater proportion of this association mediated by HOMA-IR in males. These findings have critical translational implications for developing sex-tailored public health strategies to mitigate obesity-related aging risks. Clinically, our identification of nonlinear thresholds provides potential screening targets for early intervention. In females, who demonstrate heightened susceptibility to visceral adiposity impacts even at comparable adiposity levels, preventive measures should focus on midlife adiposity monitoring with intensified lifestyle interventions during critical windows like menopausal transition. In males, for whom IR mediates over 20% of the VAI–BA relationship, clinical approaches could prioritize metabolic health optimization through IR-targeted therapies alongside visceral fat reduction. Public health initiatives could incorporate these insights through: (a) development of sex-stratified VAI thresholds in aging risk algorithms; (b) tailored community-based programs (e.g., women-focused visceral fat reduction groups, men-centered metabolic health workshops); and (c) early implementation of life-course interventions in high-risk populations identified through these biomarkers. Future longitudinal studies should validate these thresholds and test whether sex-specific reductions in visceral adiposity effectively slow progression of BA, potentially informing precision prevention frameworks for aging-related health risks.

## Supporting information

**S1 Table. Formulae for KDMAge.**
(DOCX)

**S2 Table. Definitions of clinical covariates.**
(DOCX)

**S3 Table. Mediating effects of diabetes mellitus between VAI and biological aging among the whole population.**
(DOCX)

**S4 Table. Sex-specific associations of VAI quartiles with biological aging.**
(DOCX)

**S5 Table. Sex interaction analysis.**
(DOCX)

**S6 Table. Mediation analysis of HOMA-IR in the association between VAI and biological aging.**
(DOCX)

**S7 Table. Sex interaction analysis after additional adjustment for HOMA-IR.**
(DOCX)

**S8 Table. Sex interaction analysis after additional adjustment for DM and HDL.**
(DOCX)

**S9 Table. Multivariate regression analysis after additional adjustment for DM and HDL.**
(DOCX)

**S10 Table. Threshold effect analysis after additional adjustment for DM and HDL.**
(DOCX)

**S11 Table. Mediation analysis after additional adjustment for DM and HDL.**
(DOCX)

**S12 Table. Sex interaction analysis following exclusion of DM participants.**
(DOCX)

**S13 Table. Multivariate regression analysis following exclusion of DM participants.**
(DOCX)

**S14 Table. Threshold effect analysis following exclusion of DM participants.**
(DOCX)

**S15 Table. Mediation analysis following exclusion of DM participants.**
(DOCX)

**S16 Table. Subgroup analyses of VAI–HOMA-IR associations.**
(DOCX)

**S17 Table. Subgroup analyses of HOMA-IR–BA associations.**
(DOCX)

**S18 Table. Subgroup analyses of VAI–BA associations for different race.**
(DOCX)

**S19 Table. Subgroup analyses of VAI–BA associations for different age groups.**
(DOCX)

**S1 Fig. Restricted cubic splines analyses after additional adjustment for DM and HDL.**
(DOCX)

**S2 Fig. Restricted cubic splines analyses following exclusion of DM participants.**
(DOCX)

**S1. Graphical abstract image.**
(TIF)

## Acknowledgments

This work is indebted to the NHANES program for providing essential public data resources. We particularly recognize the commitment of volunteers and researchers involved in data collection from 1999 to 2018.

## Author contributions

**Conceptualization:** Jia Yang.

**Data curation:** Jia Yang, Tiejun Liu.

**Formal analysis:** Jia Yang.

**Methodology:** Jia Yang.

**Project administration:** Haifeng Liu.

**Supervision:** Haifeng Liu.

**Validation:** Haifeng Liu, Xupeng Huang.

**Visualization:** Jia Yang.

**Writing – original draft:** Jia Yang, Haifeng Liu.

**Writing – review & editing:** Jia Yang, Haifeng Liu, Xupeng Huang, Zimin Fu, Jie Zhou, Tiejun Liu, Weimin Zhao.

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
