## [Decision Letter · Decision Letter 0]

1 Jul 2025

Dear Dr. Liu,

Thank you for submitting your manuscript to PLOS ONE. After careful consideration, we feel that it has merit but does not fully meet PLOS ONE’s publication criteria as it currently stands. Therefore, we invite you to submit a revised version of the manuscript that addresses the points raised during the review process.

We look forward to receiving your revised manuscript.

Kind regards,

Amin Mansoori

Academic Editor

PLOS ONE

Journal Requirements: 

Additional Editor Comments:

Dear Authors,

Thank you for submitting your manuscript to PLOS One. It has now been reviewed by two independent experts. While they acknowledge that your work presents novel and interesting contributions suitable for publication, they have identified several areas that require clarification and improvement.

I kindly ask you to carefully revise the manuscript, addressing all the reviewers’ comments and concerns in detail. Please submit a revised version along with a point-by-point response to the reviewers’ feedback.

Should you have any questions, please do not hesitate to contact me.

Best regards,

Amin Mansoori

Reviewers' comments:

Reviewer's Responses to Questions

**Comments to the Author**

1. Is the manuscript technically sound, and do the data support the conclusions?

Reviewer #1: Yes

Reviewer #2: Yes

2. Has the statistical analysis been performed appropriately and rigorously?

Reviewer #1: Yes

Reviewer #2: Yes

3. Have the authors made all data underlying the findings in their manuscript fully available?

Reviewer #1: Yes

Reviewer #2: Yes

4. Is the manuscript presented in an intelligible fashion and written in standard English?

Reviewer #1: Yes

Reviewer #2: Yes

Reviewer #1: Main strengths of the study:

• Relevant and under-explored study topic

• Introduction clearly setting out the theme and objectives of the study

• Methodology with a reference population representative of the U.S. population

• Well-conducted multivariate statistical analysis

• Results well presented in the form of tables of figures

• Detailed discussion contains strengths and limitations

Suggestions:

• Add in the title the target population: Cross sectional analysis of NHANES (1999-2018) data

Question:

• Why did you use VAI rather than NHANES DXA data on Visceral adipose tissue “VAT” quantity (mass, volume, or surface area) as a more direct measure of visceral adiposity ?

Reviewer #2: The original article entitled “Sex differences in the association between visceral adiposity index and biological aging: a mediation analysis of insulin resistance” was well-written and easy to understand. The study investigated the association between visceral adiposity index and biological aging, with insulin resistance playing a mediating role in homeostasis between sexes. The sample size was large, and the authors made efforts to provide guidelines for obesity and aging. However, I have some suggestion to improve the paper.

The introduction was well-written, but too long. I suggest summarizing it.

The resolution of the Figures is low

It would be better to define clinical features, including HTN, DM, CKD, CVD and Cancer.

**Do you want your identity to be public for this peer review?** For information about this choice, including consent withdrawal, please see our Privacy Policy

Reviewer #1: No

Reviewer #2: No

---

## [Author Response · Author response to Decision Letter 1]

9 Jul 2025

Dear PLOS ONE Editorial Team,

Thank you for giving us the opportunity to submit a revised draft of our manuscript titled “Sex differences in the association between visceral adiposity index and biological aging: a mediation analysis of insulin resistance” [PONE-D-25-26841R1] to PLOS One. We appreciate the time and effort that you and the reviewers have dedicated to providing valuable feedback on our manuscript. We are grateful to the reviewers for their insightful comments, which have improved the paper. We have incorporated changes that reflect all the suggestions and concerns provided by the reviewers. The reviewer comments are laid out below in italicized font and specific concerns have been numbered. Our replies appear in blue font and key points are emphasized in red. All corresponding revisions in the manuscript are marked in yellow. Furthermore, we would like to show the details as follows:

Responses to Comments of Reviewer #1:

Comments 1: Add in the title the target population: Cross sectional analysis of NHANES (1999-2018) data.

Response: We think this is an excellent suggestion. We have added the target population and database information to the title as suggested. The revised title now reads: "Sex differences in the association between visceral adiposity index and biological aging: a cross-sectional analysis of NHANES 1999–2018 with mediation by insulin resistance"

Comments 2: Why did you use VAI rather than NHANES DXA data on Visceral adipose tissue “VAT” quantity (mass, volume, or surface area) as a more direct measure of visceral adiposity?

Response: We sincerely appreciate the reviewer's insightful query regarding our selection of the visceral adiposity index (VAI) over DXA-derived visceral adipose tissue (VAT) quantification. We acknowledge the well-established value of VAT as a more direct measure of visceral adiposity, evidenced by its documented associations with diabetes mellitus [1], systemic inflammation [2], and cognitive function [3]. However, our methodological choice of VAI was based on three key considerations:

Clinical applicability and feasibility

VAI integrates routinely measured clinical parameters (waist circumference, BMI, triglycerides, HDL) into a sex-specific formula. This allows broader implementation in resource-limited settings where DXA is unavailable, aligning with our goal of developing accessible biomarkers for public health use [4]. DXA, while providing precise anatomical VAT quantification, requires specialized equipment and expertise, limiting its scalability in population-based screening.

Comprehensive pathophysiological assessment

Unlike VAT mass/volume/area, which solely reflects morphological adiposity, VAI simultaneously captures both visceral fat accumulation and associated metabolic dysfunction (dyslipidemia, insulin resistance) [5]. This dual capacity is critical because metabolic dysregulation, not fat mass alone, drives obesity-related aging mechanisms [6,7]. Our mediation results further validate VAI’s ability to reflect IR-mediated biological aging pathways.

Data availability and cohort integrity

It is important to note that DXA-derived VAT data in NHANES were systematically collected from 2011 onward. Restricting analyses to DXA-available cycles (2011–2018) would reduce the sample size by more than 60% and disproportionately exclude older adults (who are underrepresented in DXA subsamples due to the exclusion criteria of 60 years or older). This would compromise the statistical power of sex-stratified mediation analyses and introduce selection bias. VAI facilitated the optimal utilization of 20-year nationally representative data (1999–2018).

To further address your concerns about VAT, we performed the following analyses using direct measures of VAT, including mass, volume, and area, to further validate the association between visceral adiposity and biological aging (KDMAge and KDMAgeAccel risk).

The results are consistent with our primary results concerning the VAI–biological aging association. This consistency suggests that both VAT and VAI serve as outstanding indicators for evaluating the link between visceral adiposity and advanced biological aging. Nevertheless, based on the three key considerations outlined previously, we maintained VAI as the primary measure of visceral adiposity for this study. We would like to once again express our gratitude to the reviewer for the attention to this important methodological dimension, which indeed highlights valuable directions for future research in this field.

The specific analysis process is as follows: Following full adjustment for covariates, multivariable linear and logistic regression analyses demonstrated significant positive correlations between VAT and measures of biological aging (Table 1). Furthermore, restricted cubic splines revealed significant nonlinear relationships between VAT and biological aging (Fig. 1). Please refer to "Response to Reviewers.docx" for details.

References:

1.Tripathi H, Singh A, Farheen, Prakash B, Dubey DK, Sethi P, Jadon RS, Ranjan P, Vikram NK. The Metabolic Score for Visceral Fat (METS-VF) as a predictor of diabetes mellitus: Evidence from the 2011-2018 NHANES study. PLoS One. 2025 Feb 11;20(2):e0317913. doi: 10.1371/journal.pone.0317913. PMID: 39932909; PMCID: PMC11813123.

2.Liao Y, Zhou K, Lin B, Deng S, Weng B, Pan L. Associations between systemic immune-inflammatory index and visceral adipose tissue area: results of a national survey. Front Nutr. 2025 Jan 16;11:1517186. doi: 10.3389/fnut.2024.1517186. PMID: 39885869; PMCID: PMC11780491.

3.Cheng M, Meng Y, Song Z, Zhang L, Zeng Y, Zhang D, Li S. The Association Between Metabolic Score for Visceral Fat and Cognitive Function Among Older Adults in the United States. Nutrients. 2025 Jan 10;17(2):236. doi: 10.3390/nu17020236. PMID: 39861366; PMCID: PMC11768000.

4.Xue M, Zhang X, Chen K, Zheng F, Wang B, Lin Q, Zhang Z, Dong X, Niu W. Visceral adiposity index, premature mortality, and life expectancy in US adults. Lipids Health Dis. 2025 Apr 15;24(1):139. doi: 10.1186/s12944-025-02560-3. PMID: 40234930; PMCID: PMC12001622.

5.Jiang K, Luan H, Pu X, Wang M, Yin J, Gong R. Association Between Visceral Adiposity Index and Insulin Resistance: A Cross-Sectional Study Based on US Adults. Front Endocrinol (Lausanne). 2022 Jul 22;13:921067. doi: 10.3389/fendo.2022.921067. PMID: 35937809; PMCID: PMC9353944.

6.Geng J, Zhang X, Guo Y, Wen H, Guo D, Liang Q, Pu S, Wang Y, Liu M, Li Z, Hu W, Yang X, Chang P, Hu L, Li Y. Moderate-intensity interval exercise exacerbates cardiac lipotoxicity in high-fat, high-calories diet-fed mice. Nat Commun. 2025 Jan 12;16(1):613. doi: 10.1038/s41467-025-55917-8. PMID: 39800728; PMCID: PMC11725574.

7.Xia Z, Shi S, Ma X, Li F, Li X, Gaisano HY, Zhao M, Li Y, He Y, Jiang J. Mediating effect of adiponectin between free fatty acid and tumor necrosis factor-α in patients with diabetes. Nutr Diabetes. 2024 Jun 17;14(1):45. doi: 10.1038/s41387-024-00302-5. PMID: 38886355; PMCID: PMC11183252.

Responses to Comments of Reviewer #2:

Comments 1: The introduction was well-written, but too long. I suggest summarizing it.

Response: The reviewer's positive appraisal of the Introduction, characterizing it as "well-written," is deeply appreciated. In response to the suggestion to improve conciseness, we have rigorously condensed this section from 734 to 427 words (a 42% reduction). The revised version retains all critical scientific elements while enabling readers to efficiently grasp: (i) the research background, (ii) knowledge gaps, (iii) methodological framework, and (iv) clinical implications. The modified Introduction is presented below:

“The global population is rapidly aging [1]. Epidemiological projections indicate that individuals aged ≥ 60 years will constitute 16% of the global population by 2030, rising to 22% (2.1 billion) by mid-century [2]. Aging is robustly linked to increased risks of multiple chronic conditions, including diabetes mellitus (DM), cardiovascular disease (CVD), hypertension (HTN), chronic kidney disease (CKD), and cancer [3–5]. Given the profound impact of these conditions on healthcare burdens and age-related mortality, accurately assessing biological aging (BA) is crucial for developing interventions to decelerate it. Biological age, a core indicator of physiological decline, provides superior predictive value over chronological age [6]. The Klemera-Doubal method (KDMAge) quantifies BA by integrating multisystem biomarkers (e.g., metabolic, inflammatory, cardiovascular) [7]. Unlike single biomarkers, KDMAge captures individual heterogeneity in aging trajectories through multi-system interactions, serving as an effective predictor of disease risk, functional decline, and mortality [8].

The global overweight/obesity epidemic poses a major public health threat, with prevalence rising exponentially [9]. Obesity independently predicts CVD, HTN, DM, and cancer, ranking as the second leading modifiable mortality risk factor after tobacco in Western populations [10]. In the United States, 70% of adults and 19% of adolescents are affected [11]. Significant sex-based differences exist: females show higher obesity prevalence (18% vs. 14% in males in 2020) with projections indicating persistent disparities (27% vs. 23% by 2035). Adipose distribution also differs markedly, with gluteofemoral predominance in females versus abdominal in males, influenced by biological (e.g., sex hormones) and sociocultural factors [12]. Given these health implications, effective obesity assessment is critical. While body mass index (BMI) and waist circumference (WC) provide unidimensional measures, the visceral adiposity index (VAI) offers a sex-specific metric integrating morphological (BMI, WC) and biochemical markers (triglycerides, HDL). This enables simultaneous evaluation of visceral fat accumulation and metabolic dysfunction, establishing VAI as a validated clinical marker for visceral obesity [13].

Obesity accelerates aging through pathways including vascular dysfunction, insulin resistance (IR), chronic inflammation, and lifestyle factors [14,15]. Of these, IR—clinically measured by homeostatic model assessment (HOMA-IR)—is particularly critical. Obesity impairs insulin signaling via chronic inflammation, lipotoxicity, and metabolic dysregulation [16], while the synergy of obesity and IR promotes age-related comorbidities (e.g., DM, CVD) that exacerbate systemic frailty [17]. However, sex-specific mechanisms mediating the obesity–aging association remain poorly understood, particularly regarding visceral adiposity and IR pathways. Leveraging the National Health and Nutrition Examination Survey (NHANES), the purpose of our investigation was to systematically examine sex-specific associations between VAI and BA while quantifying the mediating role of IR, thereby offering novel insights to inform public health strategies aimed at mitigating obesity-related aging risks.”

Comments 2: The resolution of the Figures is low.

Response: Thanks for the feedback on the figure resolution. We agree that high-resolution figures are essential for clarity and meeting journal standards. To address this point comprehensively, we have meticulously processed all figures using the PLOS-specific Preflight Analysis and Conversion Engine (PACE) digital diagnostic tool (https://pacev2.apexcovantage.com/). This tool is designed to ensure figures meet PLOS's stringent technical requirements. Following this optimization process, all figures now achieve a resolution of 600 dpi. We are confident that this enhancement significantly improves the clarity and readability of both textual elements and graphical components within the figures. The revised high-resolution figures have been uploaded to the designated “Figures” section in the manuscript submission system. Reviewers can download the original files to verify the improvements. We believe these modifications, prompted by the reviewer's insightful comments, are crucial for fully complying with PLOS guidelines and for ensuring that readers can interpret the figures without any difficulty. We are grateful for this constructive suggestion, which has undoubtedly strengthened the presentation quality of our manuscript.

Comments 3: It would be better to define clinical features, including HTN, DM, CKD, CVD and Cancer.

Response: We are grateful to the reviewer for their insightful suggestion to elucidate the definitions of clinical covariates. In direct response, the Materials and Methods–Assessment of covariates section has been revised to include standardized diagnostic criteria based on the previous studies (lines 192–200 in the Revised Manuscript with Track Changes): HTN is defined as systolic blood pressure > 140 mmHg or diastolic blood pressure > 90 mmHg or physician diagnosis; CVD encompasses congestive heart failure, coronary artery disease, myocardial infarction, angina pectoris, or stroke; DM requires glycated hemoglobin ≥ 6.5%, fasting glucose ≥ 126 mg/dL, 2-h oral glucose tolerance test ≥ 200 mg/dL, antidiabetic use, or physician confirmation; CKD is identified by estimated glomerular filtration rate < 60 mL/min/1.73m² (CKD-EPI equation) or urine albumin-creatinine ratio > 30 mg/g; cancer is based on self-reported physician-diagnosed malignancy. S2 Table comprehensively documents disease definitions with supporting references, enhancing methodological transparency without sacrificing conciseness.

---

## [Decision Letter · Decision Letter 1]

8 Aug 2025

Dear Dr. Liu,

We look forward to receiving your revised manuscript.

Kind regards,

Amin Mansoori

Academic Editor

PLOS ONE

Journal Requirements:

Additional Editor Comments:

Dear Author,

Thank you for submitting your revised manuscript. Upon evaluation, I find that the paper has improved significantly; however, the reviewer has provided constructive comments to further enhance the manuscript. I encourage you to carefully address these suggestions and submit an updated version for final consideration.

We appreciate your efforts and look forward to receiving your revised manuscript.

Best regards,

Reviewer's Responses to Questions

**Comments to the Author**

Reviewer #1: All comments have been addressed

Reviewer #3: (No Response)

2. Is the manuscript technically sound, and do the data support the conclusions?

Reviewer #1: Yes

Reviewer #3: Partly

3. Has the statistical analysis been performed appropriately and rigorously?

Reviewer #1: Yes

Reviewer #3: Yes

4. Have the authors made all data underlying the findings in their manuscript fully available?

Reviewer #1: Yes

Reviewer #3: Yes

5. Is the manuscript presented in an intelligible fashion and written in standard English?

Reviewer #1: Yes

Reviewer #3: Yes

Reviewer #1: (No Response)

Reviewer #3: Dear Author,

I read your manuscript carefully. Your work is valuable but I have several comments to improve it:

1- Given the diverse racial/ethnic composition of the NHANES sample (e.g., Mexican American, Non-Hispanic White, Non-Hispanic Black), it would be valuable to perform subgroup or sensitivity analyses to examine whether the observed associations between VAI, biological aging, and insulin resistance differ across racial/ethnic groups. This could enhance the generalizability and interpretation of your findings.

2- While the cross-sectional design limitation is acknowledged, I recommend elaborating on its implications, particularly regarding causal inference and temporality. Discussing how reverse causation might influence the observed relationships would improve the depth of the discussion.

3- Age may act as a potential effect modifier in the relationship between VAI and biological aging. Please consider testing for an interaction between age and VAI, and if significant, stratifying the analyses accordingly. This may provide important age-related insights into the adiposity-aging relationship.

4- Given the strong association between diabetes and both insulin resistance and biological aging, a sensitivity analysis excluding individuals with diagnosed diabetes could help evaluate whether the observed associations are independent of overt diabetic status.

5-The conclusion could be expanded to highlight the potential clinical and public health implications of your findings. For example, discussing how sex-specific strategies targeting visceral adiposity may mitigate aging-related health risks would underscore the translational value of the study.

Best Regards,

**Do you want your identity to be public for this peer review?** For information about this choice, including consent withdrawal, please see our Privacy Policy

Reviewer #1: No

Reviewer #3: No

---

## [Author Response · Author response to Decision Letter 2]

10 Aug 2025

Response Letter

Dear PLOS ONE Editorial Team,

We sincerely appreciate the opportunity to submit this revised manuscript entitled ”Sex differences in the association between visceral adiposity index and biological aging: a cross-sectional analysis of NHANES 1999–2018 with mediation by insulin resistance” (PONE-D-25-26841R1) to PLOS ONE. We extend our deepest gratitude to the editorial team and reviewers for their substantial time investment and insightful critiques, which have significantly strengthened our work.

All reviewer comments have been meticulously addressed through comprehensive revisions. Below, we present the reviewers' comments in italics, followed by our point-by-point responses in blue text (with key modifications highlighted in red). Supporting documentation appears in standard font, while manuscript changes are tracked using yellow highlighting.

The detailed responses follow this section:

Responses to Comments of Reviewer #3:

Comments 1: Given the diverse racial/ethnic composition of the NHANES sample (e.g., Mexican American, Non-Hispanic White, Non-Hispanic Black), it would be valuable to perform subgroup or sensitivity analyses to examine whether the observed associations between VAI, biological aging, and insulin resistance differ across racial/ethnic groups. This could enhance the generalizability and interpretation of your findings.

Response:

We would like to express our profound gratitude to the reviewer for underscoring the significance of racial/ethnic heterogeneity in our analyses. In response, we conducted comprehensive subgroup analyses across racial/ethnic categories (Mexican American, Non-Hispanic White, Non-Hispanic Black, Other) within the whole cohort and sex-stratified subgroups. Furthermore, three key relationships were examined: (a) associations of visceral adiposity index (VAI) with homeostasis model assessment of insulin resistance (HOMA-IR); (b) relationships between HOMA-IR and biological aging (BA); and (c) associations between VAI and BA.

Key findings:

All racial/ethnic subgroups consistently demonstrated positive associations mirroring our primary results;

Significant race/ethnicity interaction effects (Pinteraction < 0.05) emerged for both VAI–HOMA-IR relationships and VAI–BA associations, indicating differential magnitudes of these positive associations across racial/ethnic subgroups.

These findings both confirm the robustness of our primary observations and reveal important population-specific variations. Notably, identifying these differential effects enhances the generalizability of our conclusions and provides valuable insights for targeted interventions. We are grateful for this insightful suggestion, which significantly strengthened our study's epidemiological relevance.

The detailed methodological description (Lines 193–201) and complete analytical results (Lines 422-435) appear in the “Revised Manuscript with Track Changes,” while relevant supporting documentation (Tables S16–S18) can be accessed in the 'Supplementary Information' file.

Revised Manuscript with Track Changes (Lines 193–201):

“To address potential effect modification by demographic factors, we conducted comprehensive subgroup and interaction analyses. Given the racial/ethnic diversity of NHANES participants (Mexican American, Non-Hispanic White, Non-Hispanic Black, Other), we (a) performed race/ethnicity-stratified analyses of VAI–IR, IR–BA and VAI–BA associations; (b) and tested multiplicative interaction terms (VAI × race/ethnicity) in fully adjusted models. Additionally, since age may modify relationships between VAI and BA, we (a) assessed age interaction effects through VAI × age terms; (b) and conducted age-stratified analyses using predefined categories (20–39, 40–59, ≥ 60 years). These analyses were implemented in both the whole cohort and sex-specific subgroups using multivariable-adjusted models. This approach enhances generalizability while identifying population-specific patterns in the visceral adiposity-aging relationship.”

Revised Manuscript with Track Changes (Lines 422–435):

“Consistent with our primary findings, stratified analyses across racial/ethnic subgroups (Mexican American, Non-Hispanic White, Non-Hispanic Black, Other) revealed persistently positive associations between VAI, BA, and IR in all populations. Three key patterns emerged: (a) VAI–HOMA-IR associations remained significantly positive across all racial/ethnic groups in the whole cohort and sex-stratified subgroups (S16 Table). Formal interaction testing indicated significant effect modification by race/ethnicity in the whole population (Pinteraction = 0.044) and among females (Pinteraction = 0.018). (b) HOMA-IR–BA relationships demonstrated consistent positive effects in all racial/ethnic subgroups without significant interaction (Pinteraction > 0.05 for whole, male, and female cohorts; S17 Table), indicating race-invariant mediation pathways. (c) The VAI–BA associations maintained significant positive effects across racial/ethnic strata (S18 Table), with significant interaction effects observed in the whole population (KDMAge: Pinteraction = 0.021; KDMAgeAccel: Pinteraction = 0.006) and among males (KDMAge: Pinteraction = 0.025; KDMAgeAccel: Pinteraction = 0.026). These findings confirm the reliability of our primary observations while identifying potential racial/ethnic differences, which enhances the generalizability of the study results and provides a basis for targeted interventions.”

Relevant supporting documentation:

S16 Table. Subgroup analyses of VAI–HOMA-IR associations.

Race/Ethnicity N (%) VAI–HOMA-IR associations P for interaction

β (95% CI) P-value

Whole population

Mexican American 3357 (17.23) 0.47 (0.31–0.63) <0.001 0.044

Non-Hispanic White 8996 (46.17) 0.40 (0.30–0.50) <0.001

Non-Hispanic Black 3803 (19.52) 1.15 (0.77–1.52) <0.001

Other 3330 (17.09) 0.33 (0.05–0.61) 0.023

Females

Mexican American 1641 (16.86) 0.54 (0.30–0.78) <0.001 0.018

Non-Hispanic White 4420 (45.42) 0.45 (0.27–0.64) <0.001

Non-Hispanic Black 1966 (20.20) 1.33 (0.76–1.90) <0.001

Other 1705 (17.52) 0.53 (0.24–0.81) <0.001

Males

Mexican American 1716 (17.59) 0.41 (0.18–0.64) <0.001 0.204

Non-Hispanic White 4576 (46.91) 0.36 (0.25–0.47) <0.001

Non-Hispanic Black 1837 (18.83) 1.01 (0.55–1.47) <0.001

Other 1625 (16.66) 0.23 (-0.06–0.52) 0.118

The models were adjusted for age, sex (only in the model of the whole population), education, marital status, poverty status, smoking status, alcohol consumption, M/VPA, HTN, CVD, cancer, and CKD. HOMA-IR, homeostasis model assessment of insulin resistance; VAI, visceral adiposity index; CI, confidence interval.

S17 Table. Subgroup analyses of HOMA-IR–BA associations.

Race/Ethnicity N (%) HOMA-IR–KDMAge associations P for interaction HOMA-IR–KDMAgeAccel associations P for interaction

β (95% CI) P-value OR (95% CI) P-value

Whole population

Mexican American 3357 (17.23) 0.20 (0.01–0.42) 0.007 0.481 1.05 (1.01–1.11) 0.011 0.609

Non-Hispanic White 8996 (46.17) 0.37 (0.28–0.47) <0.001 1.08 (1.06–1.11) <0.001

Non-Hispanic Black 3803 (19.52) 0.34 (0.22–0.46) <0.001 1.05 (1.02–1.08) 0.001

Other 3330 (17.09) 0.38 (0.17–0.58) <0.001 1.07 (1.03–1.12) 0.001

Females

Mexican American 1641 (16.86) 0.19 (-0.03–0.41) 0.091 0.479 1.06 (1.01–1.11) 0.016 0.816

Non-Hispanic White 4420 (45.42) 0.37 (0.23–0.51) <0.001 1.08 (1.04–1.12) <0.001

Non-Hispanic Black 1966 (20.20) 0.30 (0.16–0.44) <0.001 1.05 (1.01–1.09) 0.011

Other 1705 (17.52) 0.42 (0.21–0.62) <0.001 1.07 (1.02–1.13) 0.013

Males

Mexican American 1716 (17.59) 0.19 (-0.11–0.49) 0.219 0.560 1.04 (0.95–1.13) 0.407 0.732

Non-Hispanic White 4576 (46.91) 0.37 (0.24–0.50) <0.001 1.08 (1.05–1.11) <0.001

Non-Hispanic Black 1837 (18.83) 0.38 (0.17–0.59) <0.001 1.05 (1.00–1.10) 0.06

Other 1625 (16.66) 0.35 (0.13–0.57) 0.002 1.08 (1.02–1.14) 0.011

The models were adjusted for age, sex (only in the model of the whole population), education, marital status, poverty status, smoking status, alcohol consumption, M/VPA, HTN, CVD, cancer, and CKD. HOMA-IR, homeostasis model assessment of insulin resistance; KDMAge, Klemera-Doubal method age; KDMAgeAccel, KDMAge acceleration; CI, confidence interval.

S18 Table. Subgroup analyses of VAI–BA associations for different race/ethnicity.

Race/Ethnicity N (%) VAI–KDMAge associations P for interaction VAI–KDMAgeAccel associations P for interaction

β (95% CI) P-value OR (95% CI) P-value

Whole population

Mexican American 3357 (17.23) 0.64 (0.40–0.88) <0.001 0.021 1.08 (1.03–1.13) 0.002 0.006

Non-Hispanic White 8996 (46.17) 0.85 (0.68–1.01) <0.001 1.16 (1.13–1.20) <0.001

Non-Hispanic Black 3803 (19.52) 1.15 (0.64–1.66) <0.001 1.15 (1.06–1.25) 0.001

Other 3330 (17.09) 0.43 (0.15–0.71) 0.003 1.08 (1.03–1.12) <0.001

Females

Mexican American 1641 (16.86) 0.77 (0.42–1.13) <0.001 0.334 1.18 (1.08–1.28) <0.001 0.642

Non-Hispanic White 4420 (45.42) 1.04 (0.78–1.30) <0.001 1.24 (1.18–1.31) <0.001

Non-Hispanic Black 1966 (20.20) 1.48 (0.88–2.08) <0.001 1.22 (1.10–1.36) <0.001

Other 1705 (17.52) 0.92 (0.42–1.41) <0.001 1.14 (1.06–1.23) <0.001

Males

Mexican American 1716 (17.59) 0.53 (0.22–0.84) 0.001 0.025 1.04 (0.99–1.09) 0.141 0.026

Non-Hispanic White 4576 (46.91) 0.73 (0.50–0.95) <0.001 1.13 (1.08–1.18) <0.001

Non-Hispanic Black 1837 (18.83) 0.84 (0.13–1.56) 0.023 1.09 (0.99–1.20) 0.095

Other 1625 (16.66) 0.27 (0.01–0.52) 0.041 1.06 (1.02–1.11) 0.005

The models were adjusted for age, sex (only in the model of the whole population), education, marital status, poverty status, smoking status, alcohol consumption, M/VPA, HTN, CVD, cancer, and CKD. VAI, visceral adiposity index; KDMAge, Klemera-Doubal method age; KDMAgeAccel, KDMAge acceleration; CI, confidence interval.

Comments 2: While the cross-sectional design limitation is acknowledged, I recommend elaborating on its implications, particularly regarding causal inference and temporality. Discussing how reverse causation might influence the observed relationships would improve the depth of the discussion.

Response:

We are grateful to the reviewer for the thoughtful feedback regarding the limitations of our cross-sectional design. In direct response to this comment, we have substantially expanded the limitations of the Discussion section (originally 38 words → now 432 words), incorporating six additional references. Specifically, we have (a) analysed the possibility of reverse causation in the observed associations, and delineated potential biological mechanisms through which accelerated BA might drive visceral adiposity accumulation; (b) critically examined how such reverse causation could influence the interpretation of our observed associations. This revision significantly enhances the depth of our methodological discussion and provides important context for interpreting the findings. We believe these additions will allow readers to more objectively evaluate our results while strengthening the scientific rigor of our conclusions.

The revised text appears in Lines 541-576 of the “Revised Manuscript with Track Changes”.

Revised Manuscript with Track Changes (Lines 541–576):

“First, the primary limitation of this study stems from its cross-sectional design, which inherently precludes definitive causal inference regarding the temporal sequence between VAI and BA, particularly in elucidating sex-specific mechanisms. While our analyses revealed significant associations and mediation effects, this design cannot establish whether elevated VAI precedes and contributes to accelerated BA, or conversely, whether accelerated BA itself drives increases in VAI. Compelling evidence suggests that age-related lipid turnover decline and adipose tissue dysfunction may constitute reverse causal pathways. Specifically, longitudinal data demonstrate that adipose lipid turnover decreases by approximately 4.7% annually with aging, disrupting the dynamic equilibrium of lipid storage and clearance, even under constant caloric intake, potentially driving weight gain exceeding 20% over decades [55]. This dysregulation is particularly pronounced in visceral adipose tissue due to diminished preadipocyte differentiation capacity, reduced lipolytic enzyme activity (e.g., hormone-sensitive lipase), and blunted catecholamine responsiveness, collectively impairing lipid mobilization. Notably, the recent identification of CP-A adipose progenitor cells, a subpopulation emerging during middle age that exhibits hyperactive adipogenic potential through LIFR-STAT3 signaling activation, may further explain age-dependent visceral adipose tissue expansion [56]. Concurrently, adipose tissue senescence is characterized by chronic low-grade inflammation involving increased immune cell infiltration (e.g., TNF-α, IL-6), tissue fibrosis, and adipokine dysregulation (e.g., reduced adiponectin, elevated leptin), establishing a self-perpetuating vicious cycle whereby aging promotes adipose dysfunction that subsequently exacerbates systemic metabolic deterioration and accelerates organismal decline [57]. Additional contributors to age-associated visceral adiposity include muscle mass loss, progressive reductions in physical activity levels, and hormonal alterations [11,58,59]. Critically, the observed mediation proportion of HOMA-IR, while statistically significant, must also be interpreted cautiously within this bidirectional framework, as aging-related metabolic dysfunction may simultaneously elevate HOMA-IR independently of prior adiposity changes [60]. Importantly, this potential reverse causation may influence our key finding of stronger VAI–BA associations in females. Estrogen, predominantly in premenopausal women, maintains subcutaneous fat distribution and functionality while delaying visceral adipose tissue accumulation through ERα-mediated mechanisms; however, the abrupt estrogen decline during menopause may precipitate rapid visceral adipogenesis, potentially amplifying the observed VAI–BA relationship in female cohorts [59]. Furthermore, the biphasic nonlinear associations and threshold effects we identified may reflect shifting dominance between causal directions: below VAI thresholds, traditional “adiposity-driven aging” mechanisms (lipotoxicity, inflammation) likely predominate, whereas above thresholds, aging-induced metabolic dysregulation may progressively disrupt adipose homeostasis, creating a self-reinforcing pathological loop [57]. Consequently, the strength and sex-specific patterns of the associations reported here require validation in longitudinal studies tracking temporal dynamics of VAI, HOMA-IR, and BA biomarkers to establish precedence. Future experimental interventions targeting visceral adiposity reduction would provide stronger evidence for causal pathways and their sexual dimorphism.”

References:

11. Nunan E, Wright CL, Semola OA, Subramanian M, Balasubramanian P, Lovern PC, et al. Obesity as a premature aging phenotype — implications for sarcopenic obesity. GeroScience. 2022;44: 1393–1405. doi:10.1007/s11357-022-00567-7

55. Arner P, Bernard S, Appelsved L, Fu K-Y, Andersson DP, Salehpour M, et al. Adipose lipid turnover and long-term changes in body weight. Nat Med. 2019;25: 1385–1389. doi:10.1038/s41591-019-0565-5

56. Wang G, Li G, Song A, Zhao Y, Yu J, Wang Y, et al. Distinct adipose progenitor cells emerging with age drive active adipogenesis. Science. 2025;388: eadj0430. doi:10.1126/science.adj0430

57. Nguyen TT, Corvera S. Adipose tissue as a linchpin of organismal ageing. Nat Metab. 2024;6: 793–807. doi:10.1038/s42255-024-01046-3

58. Winters-van Eekelen E, van der Velde JHPM, Boone SC, Westgate K, Brage S, Lamb HJ, et al. Objectively Measured Physical Activity and Body Fatness: Associations with Total Body Fat, Visceral Fat, and Liver Fat. Med Sci Sports Exerc. 2021;53: 2309–2317. doi:10.1249/MSS.0000000000002712

59. Vecchiatto B, Castro TL, Ferreira NJR, Evangelista FS. Healthy adipose tissue after menopause: contribution of balanced diet and physical exercise. Explor Endocr Metab Dis. 2025; 101424. doi:10.37349/eemd.2025.101424

60. Palmer AK, Jensen MD. Metabolic changes in aging humans: current evidence and therapeutic strategies. J Clin Invest. 132: e158451. doi:10.1172/J

---

## [Decision Letter · Decision Letter 2]

15 Sep 2025

Sex differences in the association between visceral adiposity index and biological aging: a cross-sectional analysis of NHANES 1999–2018 with mediation by insulin resistance

PONE-D-25-26841R2

Dear Dr. Liu,

We’re pleased to inform you that your manuscript has been judged scientifically suitable for publication and will be formally accepted for publication once it meets all outstanding technical requirements.

Kind regards,

Amin Mansoori

Academic Editor

PLOS ONE

Additional Editor Comments (optional):

Dear Author,

We are pleased to inform you that your manuscript has been accepted for publication in PLOS ONE. The revisions have significantly improved the manuscript, and the current version meets the journal’s standards for publication.

Best Regards

Reviewers' comments:

Reviewer's Responses to Questions

**Comments to the Author**

Reviewer #3: All comments have been addressed

2. Is the manuscript technically sound, and do the data support the conclusions?

Reviewer #3: Yes

3. Has the statistical analysis been performed appropriately and rigorously?

Reviewer #3: Yes

4. Have the authors made all data underlying the findings in their manuscript fully available?

Reviewer #3: Yes

5. Is the manuscript presented in an intelligible fashion and written in standard English?

Reviewer #3: Yes

Reviewer #3: (No Response)

**Do you want your identity to be public for this peer review?** For information about this choice, including consent withdrawal, please see our Privacy Policy

Reviewer #3: No

---

## [Editor Report · Acceptance letter]

PONE-D-25-26841R2

PLOS ONE

Dear Dr. Liu,

I'm pleased to inform you that your manuscript has been deemed suitable for publication in PLOS ONE. Congratulations! Your manuscript is now being handed over to our production team.

Kind regards,

on behalf of

Dr. Amin Mansoori

Academic Editor

PLOS ONE